# Modeling assortative mating and genetic similarities between partners, siblings, and in-laws

Fartein Ask Torvik [1,2 ✉], Espen Moen Eilertsen[1,2], Laurie J. Hannigan[3,4,5], Rosa Cheesman[2,6], Laurence J. Howe [4], Per Magnus[1], Ted Reichborn-Kjennerud[5,7], Ole A. Andreassen [7,8], Pål R. Njølstad [9,10], Alexandra Havdahl[2,3,5,11] & Eivind Ystrom [2,5,11]

Assortative mating on heritable traits can have implications for the genetic resemblance between siblings and in-laws in succeeding generations. We studied polygenic scores and phenotypic data from pairs of partners ($n = 26{,}681$), siblings ($n = 2{,}170$), siblings-in-law ($n = 3{,}905$), and co-siblings-in-law ($n = 1{,}763$) in the Norwegian Mother, Father and Child Cohort Study. Using structural equation models, we estimated associations between measurement error-free latent genetic and phenotypic variables. We found evidence of genetic similarity between partners for educational attainment ($r_g = 0.37$), height ($r_g = 0.13$), and depression ($r_g = 0.08$). Common genetic variants associated with educational attainment correlated between siblings above 0.50 ($r_g = 0.68$) and between siblings-in-law ($r_g = 0.25$) and co-siblings-in-law ($r_g = 0.09$). Indirect assortment on secondary traits accounted for partner similarity in education and depression, but not in height. Comparisons between the genetic similarities of partners and siblings indicated that genetic variances were in intergenerational equilibrium. This study shows genetic similarities between extended family members and that assortative mating has taken place for several generations.

[1] Centre for Fertility and Health, Norwegian Institute of Public Health, Oslo, Norway. [2] PROMENTA Research Center, Department of Psychology, University of Oslo, Oslo, Norway. [3] Nic Waals Institute, Lovisenberg Diaconal Hospital, Oslo, Norway. [4] Medical Research Council Integrative Epidemiology Unit, Population Health Sciences, University of Bristol, Brisol, United Kingdom. [5] Department of Mental Disorders, Norwegian Institute of Public Health, Oslo, Norway. [6] Social, Genetic & Developmental Psychiatry Centre, Institute of Psychiatry, Psychology & Neuroscience, King's College London, London, United Kingdom. [7] Institute of Clinical Medicine, University of Oslo, Oslo, Norway. [8] NORMENT, Division of Mental Health and Addiction, Oslo University Hospital, Oslo, Norway. [9] Center for Diabetes Research, Department of Clinical Science, University of Bergen, Bergen, Norway. [10] Children and Youth Clinic, Haukeland University Hospital, Bergen, Norway. [11] These authors jointly supervised: Alexandra Havdahl, Eivind Ystrom. ✉email: fartein.ask.torvik@fhi.no

Although humans put considerable effort into finding suitable mates, genetic studies typically assume that mating is random[1–3]. However, partners resemble each other for many traits, including educational attainment[4–6], height[6–8], and psychopathology[9,10]. The selection of partners based on similarity is known as assortative mating. Partner similarity can have genetic consequences in the following generations if the assortment is based on heritable traits[6,11], which almost all human traits are[12,13]. When offspring inherit genetic variants from both parents that deviate in the same direction from the population mean, otherwise independent genetic variants can become correlated (gametic phase disequilibrium). This results in the elevated resemblance between siblings and increased genetic variation between families[11,14,15]. Increased genetic variation translates into larger variation between individuals in phenotypic expression. Assortment based on educational attainment may have particularly broad consequences. It could pose a societal challenge by concentrating human and economic resources[16] and could present a health challenge because genetic influences on educational attainment correlate with most health phenotypes[17]. Furthermore, educational attainment has increased massively over the last few generations[18]. Therefore, the genetic consequences may not yet have fully unfolded. Studies of assortative mating are needed to increase our understanding of social inequalities.

There is evidence for substantial correlations between partners at trait-associated genetic loci for educational attainment ($r = 0.65$) and height ($r = 0.20$)[6]. Assortative mating in previous generations can also be detected in samples of genomes from unrelated individuals by estimating covariance between trait-associated loci in distant parts of the genome[15]. Previous studies on assortative mating have mainly analyzed partners only. Another source of evidence for assortative mating—as yet little exploited in the literature – is systematic inflation of sibling correlations. Genetic correlations between siblings are expected to be 0.50 under random mating but could increase under assortment in previous generations. When the assortment level is stable, the sibling correlation and the genetic variance increase for each generation until an equilibrium is reached[14,19]. Simultaneous use of partner and sibling correlations can be informative regarding whether this equilibrium has been reached or whether one can expect increasing genetic variation in the future.

In-laws are another valuable set of relationships that provide information on assortative mating. Assortative mating should induce resemblance, both phenotypic and genetic, between siblings-in-law (siblings of partners or partners of siblings) and by extension between co-siblings-in-law, who are the respective partners of siblings. For brevity, we refer to these relationship types as in-laws and co-in-laws, respectively. In-laws and co-in-laws are only indirectly related, and are therefore informative for understanding the mechanisms leading to the resemblance between the partners that connect them. With direct assortment (also called primary phenotypic assortment), the similarity between partners results from assortment based on the phenotype in question. This can be distinguished from indirect assortment (also called secondary assortment), where partners resemble each other because they assort on one or more traits associated with the trait of primary interest[20–22]. An example of direct assortment would be partner selection based on observed height. An example of indirect assortment would be partner similarity in education resulting from the assortment on a secondary trait such as cognitive abilities. Under direct assortment, the expected correlations between in-laws or co-in-laws correspond to the product of the relations that connect them. Under indirect assortment, all correlations between relatives should be equally deflated. Data from

in-laws and co-in-laws can therefore be used to separate between direct and indirect assortment. In addition to providing information on mechanisms, the genetic resemblance between in-laws is a predictable but perhaps surprising phenomenon that displays consequences of assortment. A few studies, primarily using twin samples, have studied in-laws to distinguish between mechanisms of assortment[7,21,23,24]. We are not aware of any pre-existing study demonstrating resemblance between in-laws and co-in-laws for measured genetic variants.

One way to investigate genetic resemblance between individuals is to calculate correlations between their polygenic scores. A polygenic score summarizes an individual's genetic predisposition to a trait across many single nucleotide polymorphisms (SNPs)[25,26]. Polygenic scores correlate with a trait to the degree that the trait is heritable, and the polygenic scores capture that heritable component. Polygenic scores capture the heritable component when they include SNPs in linkage disequilibrium with causal variants weighted according to the true associations with the trait in the target sample. The variance in a trait explained by polygenic scores is typically lower than its heritability, indicating that polygenic scores usually do not fully succeed in capturing the heritable component[27]. As long as polygenic scores are imperfectly correlated with the genetic predispositions to a trait, the correlation between multiple individuals' polygenic scores will likely be lower than the correlations between the true set of trait-associated genetic variants. For instance, correlations between spouses' polygenic scores could be underestimated if one partner has many trait-increasing alleles included in the polygenic score whereas the other has many trait-increasing alleles not included in the polygenic score. Polygenic scores for educational attainment have been found to correlate lower in partners, at only $r = 0.18$[28] or $r = 0.11$[29], than adjusted estimates of genetic correlations based on genetic predictors and the phenotypes of partners ($r = 0.65$)[6]. Several approaches to similar scaling issues have been proposed[30,31], including structural equation modelling[32].

In the present study, we introduce a model, the Correlation in Genetic Signals (rGenSi) model, and demonstrate that phenotypic data and polygenic scores from relatives can be used to estimate signal versus noise in polygenic scores, account for the noise, and estimate the correlation between their genetic signals. We estimate the phenotypic and genetic similarity between 26,681 pairs of spouses, 2170 pair of siblings, 3905 pair of in-laws, and 1763 pairs of co-in-laws participating in the population-based Norwegian Mother, Father, and Child Cohort Study (MoBa). We investigate assortment on educational attainment, height, and depression (symptoms of major depressive disorder), which are phenotypes differentially influenced by genetic and environmental factors and with varying levels of partner similarity. We find genetic similarity between partners in all three traits, elevated (>0.50) genetic correlations between siblings, and that the genetic similarity extends to in-laws and co-in-laws. The inclusion of in-laws also allows us to separate between direct assortment on the observed traits and indirect assortment on secondary traits. We show that there is indirect assortment on secondary traits for education and depression, whereas direct assortment underlies partner similarity in height. Because siblings provide information on the level of assortment in previous generations and partners provide information on the level of assortment in the current generation, we test whether the results are consistent with stable levels of genetic assortment across generations. The results suggest no deviations from intergenerational equilibrium, indicating that assortment on these traits has been going on for at least five generations and that one should not expect further genetic consequences for succeeding generations with the present level of assortment.

## Results

**Phenotypic and polygenic similarities between relatives**. Figure 1 shows the relationship types that we study. Data on educational attainment at age 30 were gathered from governmental registers, whereas height and symptoms of depression were self-reported at the start of the study (women: mean age 30.62, SD = 4.66; men: mean age 33.18, SD = 5.37). Table 1 presents an overview of correlations between partners, siblings, in-laws, and co-in-laws in phenotype, polygenic score, and across phenotype and polygenic score. We consider associations to be significantly different from 0.00 or 0.50 when the 95% confidence intervals do not include these numbers. For all three phenotypes, there were positive correlations between partners, siblings, in-laws, and co-in-laws. All three polygenic scores were significantly associated with the individuals' own phenotype, the phenotype of their sibling, and the phenotype of their partner. For educational attainment, the polygenic score was also associated with in-laws' ($r = 0.11$, 95% CI 0.09, 0.13) and co-in-laws' ($r = 0.08$, 95% CI 0.05, 0.11) educational attainment. The polygenic scores were correlated between partners for educational attainment ($r = 0.11$, 95% CI 0.10, 0.12) and height ($r = 0.05$, 95% CI 0.03, 0.06) but not depression ($r = 0.00$, 95% CI −0.01, 0.01). The polygenic

scores were also correlated between in-laws for educational attainment ($r = 0.06$, 95% CI 0.03, 0.09) and height ($r = 0.03$, 95% CI 0.00, 0.05), and between co-in-laws for educational attainment ($r = 0.06$, 95% CI 0.01, 0.10).

**Direct versus indirect assortment**. Figure 2 shows the full structural equation model. The parameter $a$ measures the association between the observed phenotype and the latent phenotype that is the basis of partner selection. To the extent that $a < 1$, this is interpretable as evidence of indirect assortment operating alongside direct assortment. The results of fitting to the data rGenSi models are presented in Supplementary Note 1. Results with $a$ either freely estimated or fixed to 1.00 are presented in Supplementary Table 5. The estimated parameters of these different versions of the model are presented in Supplementary Table 6. Figure 3 shows the parameter estimates for the rGenSi models with the best fit for each phenotype. Partner similarity in educational attainment ($a = 0.77$, 95% CI 0.75, 0.78) and depression ($a = 0.80$, 95% CI 0.69, 0.91) appeared to result from indirect assortment on secondary phenotypes highly correlated with, but not identical to these primary phenotypes. Partner similarity in height resulted from assortment directly based on observed height ($a$ estimated at 0.97, 95% CI 0.92, 1.02, fixed to $a = 1$ in the best fitting model).

**Effects of shared environment**. The parameter $c$ measures the influence of environments that siblings shared. Supplementary Tables 5 and 6 show results with this parameter either freely estimated or fixed to 0.00. Shared environmental factors influenced educational attainment. This model had wider confidence intervals for heritability ($h^2$) and genetic signal ($s^2$) than the model without $c$, but better fit. The shared environment did not influence height and depression ($c$ fixed to 0 in the best-fitting models).

**Correlation in genetic signal between relatives**. Figure 4 illustrates the genetic correlations between relatives when estimated as correlations between polygenic scores and with the rGenSi model. For educational attainment, the latent genetic factors correlated 0.37 (95% CI 0.21, 0.67) between partners, which was

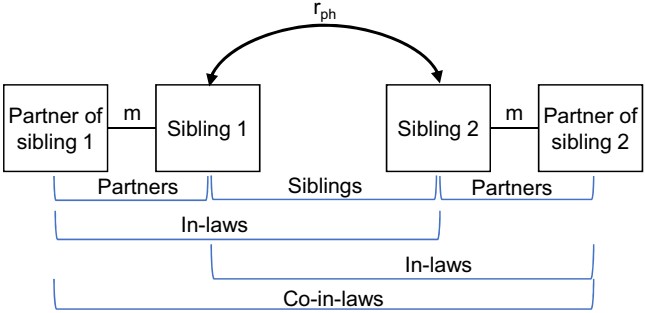

**Fig. 1 Relationship terminology.** We study four types of relationships, partners, siblings, siblings-in-law, and co-siblings-in-law (the respective partners of siblings). m = assortative mating, $r_{ph}$ = similarity between siblings. This is a conceptual model; it is not fitted to data.

**Table 1 Phenotypic and genetic correlations between partners, siblings, in-laws, and co-in-laws.**

| Relation | r(phenotype) | r(phenotype, polygenic score) | r(polygenic score) | r(genetic signal), rGenSi model |
|---|---|---|---|---|
| **Educational attainment** | | | | |
| Within individual | 1.00 | 0.28 [0.28, 0.29] | 1.00 | |
| Partner | 0.42 [0.41, 0.42] | 0.18 [0.17, 0.19] | 0.11 [0.10, 0.12] | 0.37 [0.21, 0.67] |
| Sibling | 0.39 [0.38, 0.41] | 0.20 [0.17, 0.23] | 0.55 [0.52, 0.57] | 0.68 [0.61, 0.75] |
| In-law | 0.27 [0.26, 0.28] | 0.11 [0.09, 0.13] | 0.06 [0.03, 0.09] | 0.25 [0.14, 0.46] |
| Co-in-law | 0.19 [0.17, 0.21] | 0.08 [0.05, 0.11] | 0.06 [0.01, 0.10] | 0.09 [0.03, 0.31] |
| **Height** | | | | |
| Within individual | 1.00 | 0.52 [0.51, 0.53] | 1.00 | |
| Partner | 0.16 [0.15, 0.17] | 0.09 [0.08, 0.11] | 0.05 [0.03, 0.06] | 0.13 [0.11, 0.15] |
| Sibling | 0.49 [0.47, 0.51] | 0.31 [0.28, 0.34] | 0.52 [0.49, 0.55] | 0.55 [0.50, 0.59] |
| In-law | 0.08 [0.06, 0.10] | 0.02 [−0.01, 0.04] | 0.03 [0.00, 0.06] | 0.07 [0.06, 0.08] |
| Co-in-law | 0.03 [0.00, 0.06] | 0.02 [−0.02, 0.05] | 0.02 [−0.02, 0.07] | 0.01 [0.01, 0.01] |
| **Depression** | | | | |
| Within individual | 1.00 | 0.11 [0.10, 0.12] | 1.00 | |
| Partner | 0.18 [0.17, 0.18] | 0.03 [0.02, 0.04] | 0.00 [−0.01, 0.01] | 0.08 [0.03, 0.11] |
| Sibling | 0.14 [0.12, 0.15] | 0.08 [0.05, 0.11] | 0.53 [0.50, 0.56] | 0.70 [0.48, 0.89] |
| In-law | 0.05 [0.03, 0.06] | 0.01 [−0.01, 0.03] | 0.00 [−0.03, 0.03] | 0.05 [0.03, 0.10] |
| Co-in-law | 0.04 [0.02, 0.06] | 0.01 [−0.02, 0.05] | 0.01 [−0.04, 0.06] | 0.00 [0.00, 0.01] |

Note: The first column shows phenotypic Pearson correlations between partners, siblings, in-laws, and co-in-laws; the second column shows Pearson correlations between the phenotype in one relative and the polygenic score in another; the third column shows Pearson correlations of the polygenic scores in different individuals; the fourth column shows similarity in trait-associated genetic factors estimated with the rGenSi model to adjust for noise in the assessment of polygenic scores and phenotypes. Based on 26,681 pairs of partners, 2170 pairs of siblings, 3905 pairs of siblings-in-laws, and 1763 co-siblings-in-law.

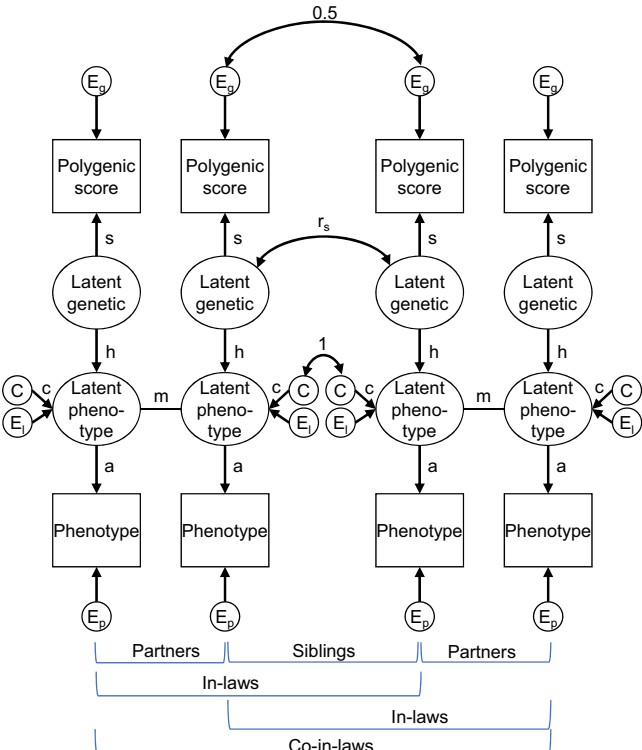

**Fig. 2 The Correlation in Genetic Signal (rGenSi) model.** Triangles represent observed variables and ovals represent latent variables. Correlations in genetic signals between relatives can be estimated by following path tracing rules. $h$ = influence of latent genetic factor on phenotype, heritability when squared; $s$ = influence of latent genetic factor on polygenic score, genetic signal; $m$ = partner similarity in mated phenotype; $a$ = association between mated and measured phenotype; $r_s$ = correlation between siblings' latent genetic factors; $C$ = shared environmental variance; $c$ = influences of shared environment; $Eg$ = residual variance of polygenic score (not related to latent genetic factor); $El$ = residual variance of latent phenotype (not related to latent genetic factor); $Ep$ = residual variance of phenotype (not related to mated phenotype). A restricted version of the model can be estimated by constraining $c$ to 0.00 and $a$ to 1.00; otherwise, these can be freely estimated. Variances are fixed to 1. The residual correlation between siblings' polygenic scores (here fixed to 0.50) can be freely estimated if influences of C are fixed to 0.00.

considerably higher than the correlation between polygenic scores ($r = 0.11$, 95% CI 0.10, 0.12). Genetic correlations were also found for in-laws ($r = 0.25$, 95% CI 0.14, 0.46) and co-in-laws ($r = 0.09$, 95% CI 0.03, 0.31). For height, the partner correlation was estimated at 0.13 (95% CI 0.11, 0.15), the in-law correlation at 0.07 (95% CI 0.06, 0.08), and the co-in-law correlation at 0.01 (95% CI 0.01, 0.01). For depression, the partner correlation was estimated at 0.08 (95% CI 0.03, 0.11), the in-law correlation at 0.05 (95% CI 0.03, 0.10), and the co-in-law correlation at 0.00 (95% CI 0.00, 0.01). Genetic correlations between siblings were estimated above expectation (>0.50) for all three phenotypes, but only for educational attainment ($r = 0.68$, 95% CI 0.61, 0.75) did the confidence interval exclude 0.50. Results were similar when using alternative p-value cut-offs for including SNPs in the polygenic scores (see Supplementary Note 2, Supplementary Figs. 7 and 8, Supplementary Tables 7, 8, and 9).

**Testing intergenerational equilibrium.** Correlations between partners in the genetic signal result from mating in the present generation, and correlations between siblings in the genetic signal result from mating in previous generations. Figure 4 shows

genetic correlations predicted across relationship types as expected in intergenerational equilibrium. For all three phenotypes, the predicted correlations matched well with the observed correlations, and models with the equilibrium constraint fit well (for educational attainment Δ-2LL 0.00, ΔDF = 1, $p = 0.987$; for height Δ-2LL = 0.40, ΔDF = 1, $p = 0.528$; for depression Δ-2LL = 1.89, ΔDF = 1, $p = 0.170$). Similarly, the correlations between the polygenic scores also fit well with the intergenerational equilibrium, with no deviations detected at the 5% significance level (for educational attainment Δ-2LL = 0.55, ΔDF = 1, $p = 0.457$; for height Δ-2LL = 0.01, ΔDF = 1, $p = 0.935$; for depression Δ-2LL = 2.78, ΔDF = 1, $p = 0.095$).

## Discussion

We found genetic similarity between partners for educational attainment, height, and depression and elevated genetic correlations between siblings. The genetic similarity extended to in-laws and co-in-laws. The partner similarity was particularly high for educational attainment, which resulted from an assortment on traits correlated with educational attainment. No deviations from intergenerational equilibrium were found in the level of genetic assortment in the three traits.

All three polygenic scores correlated with the phenotype of partners and indicated correlations between partners' genetic factors. For educational attainment, the correlations between partners' polygenic scores were close to the results of the previous studies[5,28,29]. The rGenSi adjusted genetic partner correlation was lower than Robinson et al.'s[6] (0.37 vs 0.65), but the confidence intervals overlapped with their estimate. High correlations between partners in genetic factors for educational attainment have now been found using several independent methods and in different populations. This strengthens our confidence that the results reflect the actual similarity between partners on common genetic variants associated with educational attainment. The genetic correlation of 0.37 is higher than what would be expected among half-siblings (0.25) under no assortment and can induce bias in genetic studies of educational attainment if not accounted for[1–3]. For height, the rGenSi genetic partner correlation was consistent with Robinson et al.'s[6] results from a different population. For depression, we have less basis for direct comparison of genetic partner correlations, but previous studies indicate low assortment on this trait[5,15]. The observed resemblance between in-laws or co-in-laws in measured genetic factors for educational attainment was theoretically expected and implicit in quantitative genetic models of siblings and their partners. Confirming that they exist by using genomic data aligns with our expectations and highlights the wide implications of assortative mating. This implies that health and resources are concentrated not only at the couple level but also in wider family networks. The mating within subgroups of the population is an example of endogamy that can concentrate resources and maintain social inequality.

Our results favored models where partner similarity in educational attainment and depression was accounted for by indirect assortment on secondary traits. For partner similarity in height, our results were consistent with partner selection directly on manifest height. This means that "tall people choose other tall people because they are tall"[33] (p. 382) and that there is nothing more to partner similarity in height. The direct assortment on height is in line with previous genetic studies[6]. For educational attainment and depression, the story is more complex. The association between observed education and the latent mated phenotype was not perfect, indicating that other traits than the observed played a role. The results for educational attainment are in line with previous studies finding that partner similarity in education results from the indirect assortment on correlated

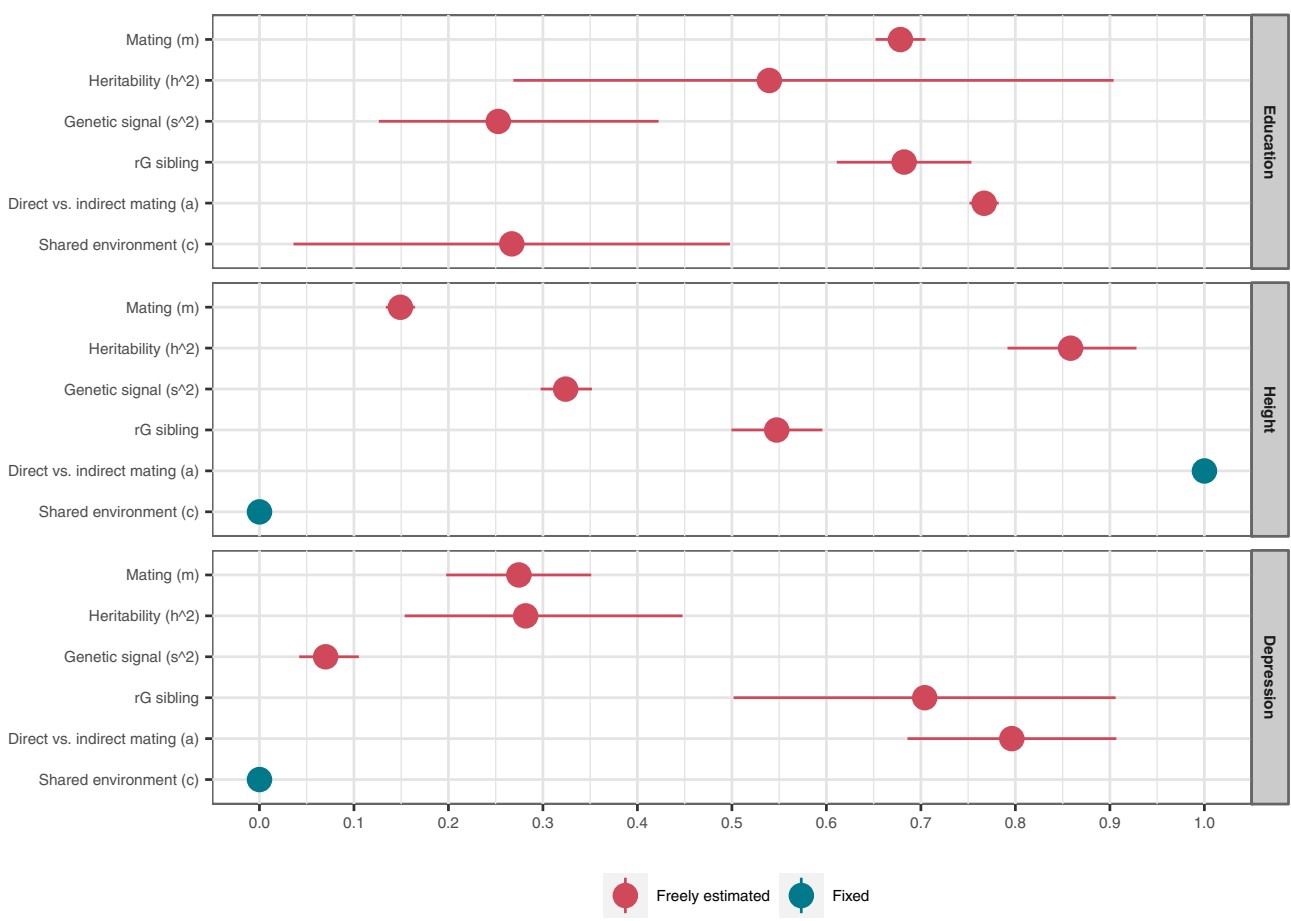

**Fig. 3 Parameter estimates.** Estimates from the best fitting rGenSi models, including 95% likelihood-based confidence intervals. Values in red are freely estimated and can take any value between 0.00 and 1.00. Values in blue are fixed ($a = 1$ or $c = 0$) in the best fitting models. Exact numbers and freely estimated values for all parameters are available in Supplementary Table 6. For height and depression, there are no effects of shared environment ($c = 0$). For height, mating is fully based on the measured height ($a = 1$), whereas for education and depression, couples assort on a correlated phenotype. rG sibling = correlation between siblings' latent genetic factors. Based on genotype data from $n = 26,681$ complete pairs of partners, $n = 2,170$ complete sibling pairs, $n = 3,905$ complete in-law pairs, and $n = 1,763$ complete co-in-law pairs, and phenotype data from $n = 63,781$ complete pairs of partners, $n = 13,455$ complete sibling pairs, $n = 21,496$ complete in-law pairs, and $n = 8,699$ complete co-in-laws pairs.

traits[6,34]. They are also consistent with reports of social homogamy in twin samples[35] because social homogamy and indirect assortment can be indistinguishable in phenotypic data[20]. If unobserved secondary traits underlie the partner similarity, individuals with highly educated families are likely to find highly educated partners even if they do not have high education themselves. Our results cannot tell us what these secondary traits might be, but plausible candidates for educational attainment include cognitive abilities, conscientiousness, other indicators of academic abilities[36,37], and the obtained education itself; most likely a combination of these. This could be clarified by investigating an assortment on these traits in tandem with educational attainment. Our depression measure was intended to assess lifetime history of major depression but might more strongly capture recent than past episodes[38]. It therefore seems plausible that factors underlying the observed partner similarity in depression could include symptoms at the time of couple formation rather than the time of observation, or phenotypes associated with depression, such as neuroticism, other mental disorders, or a general risk of psychopathology[39,40].

A genetic partner correlation can lead to increased genetic variation in the next generation and a sibling correlation elevated above 0.50. This is exactly what we found for educational attainment, with a genetic sibling correlation of 0.68 (95% CI

0.61, 0.75). This indicates that the participants' parents were also engaged in assortative mating, a finding that is in line with previous findings of correlations between trait-associated alleles in different parts of the genome[15]. Higher education has been widely available in Norway for only a few generations, but people may have always selected partners based on traits that are correlated with educational attainment today. Our results on direct versus indirect assortment indicate that such factors were important, meaning that assortment on genetic factors correlated with educational attainment may be older than assortment on education itself. The genetic partner correlations could be predicted from the genetic sibling correlations and vice versa. The results thus indicate that an assortment on genetic factors correlated with educational attainment has occurred long enough for it to be in or close to equilibrium. Results from the polygenic scores and the rGenSi model both supported this. The genetic variance increases fastest in the first generations with assortment. Genetic variance close to equilibrium can be observed after only five generations, whereas 60% of the increase in variance is seen after two generations (see Supplementary Figure 6). Hence, one could observe approximate equilibrium if assortment on variables associated with educational attainment started as late as the end of the 19th century. The larger part of the distance to equilibrium can also be covered for even newer phenomena. Our finding of genetic

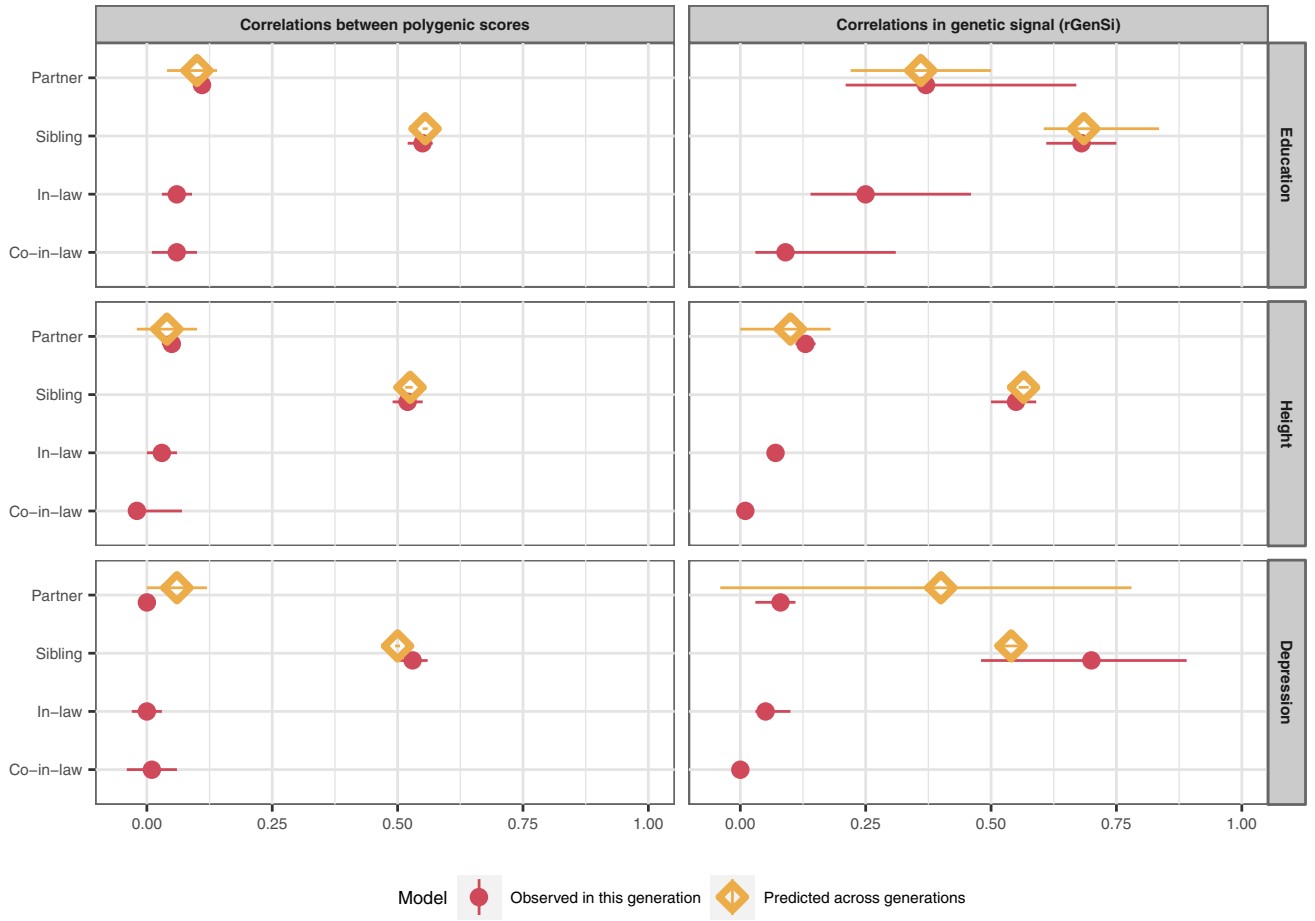

**Fig. 4 Genetic resemblance between relatives.** Correlations between partners', siblings', in-laws', and co-in-laws' genetic dispositions. Correlations were estimated as Pearson's correlations between polygenic scores (left) and as correlations between the latent genetic signals with the rGenSi model (right), including 95% likelihood-based confidence intervals. Genetic correlations from the best-fitting models. Values in red are estimated in the current generation. Values in yellow show results when partner correlations are used to predict sibling correlations and vice versa, assuming equilibrium. Predicted and observed correlations are expected to match in equilibrium. Based on genotype data from $n = 26,681$ complete pairs of partners, $n = 2,170$ complete sibling pairs, $n = 3,905$ complete in-law pairs, and $n = 1,763$ complete co-in-law pairs, and phenotype data from $n = 63,781$ complete pairs of partners, $n = 13,455$ complete sibling pairs, $n = 21,496$ complete in-law pairs, and $n = 8,699$ complete co-in-laws pairs.

equilibrium in educational attainment could therefore be expected from century-old descriptions of partner similarity in related traits[8,10] and aligns with stable genetic spouse similarity among individuals born in the first half of the 20th century[5]. It does, however, contrast with Kong et al.'s[34] finding that genetic educational assortment is a recent phenomenon in Iceland. The educational system in Iceland was developed later than in Norway, from which our sample comes. Unlike Norway, most areas of Iceland had few or no schools in the 19th century[41] and the first university was founded 100 years later than the first in Norway (1911 vs. 1811). Hence, it is possible that education became relevant as an assortment factor later in Iceland than in Norway. A weakness with this explanation is that Icelanders could have chosen partners based on factors that are related to education today even with low access to formal education. Rurality and unidentified cultural differences are therefore alternative explanations. In addition, we cannot exclude that the contrast in results is related to differences in methods, such as our use of siblings and in-laws rather than only partners. This finding therefore calls for further replication. Alternative versions of our model, presented in the Supplementary Table 5, indicated deviations from equilibrium, but those versions of the model relied on assuming direct assortment and had poor fit. If our

results are correct, we do not expect changes in the genetic variance of educational attainment[19] and associated health outcomes[17] unless the level of assortment changes. For height and depression, the results also did not indicate deviations from intergenerational equilibrium.

We used a structural equation model to estimate the correlation between partners and relatives in their genetic signal, or underlying genetic factors, for a trait. This approach is computationally efficient and can be applied in cases where only polygenic scores are available but not raw genomic data. Nevertheless, simulations showed that the model could reconstruct full genetic correlations, as well as other parameters. Correcting for measurement error is one of the main advantages of using structural equation modelling. Correcting polygenic scores for measurement error to extract the genetic signal is not yet common practice, but approaches for estimating latent genetic effects have been described[32,42]. This allowed us to scale up the genetic correlations to infer associations between the full set of genetic variants giving rise to the phenotypes rather than between the polygenic scores. The model disentangles the proportion of genetic signal ($s^2$) in the polygenic scores and may be informative about the comprehensiveness of the polygenic score as an indicator of genetic risk in the target sample. Because of this, the results were consistent

across different versions of the polygenic scores. Ideally, the latent genetic factor should be modelled as the sum of two components —trait-increasing alleles included in the polygenic score (A) plus trait-increasing alleles not included in the polygenic (B) score[34]. The polygenic score itself should then be modelled as the sum of trait-increasing alleles (A) and measurement error. This would have enabled us to distinguish measurement error from partialness of the polygenic scores. However, there were not enough degrees of freedom in the present data, and estimating the resulting correlations was sufficient to reach our objectives. Estimating direct versus indirect mating and shared environmental influences provides a more comprehensive understanding of the processes that give rise to phenotypic and genetic similarity. The inclusion of shared environment did, however, increase the width of the confidence intervals. Structural equation models can flexibly be adapted. We therefore believe that variations of the model can be applied to other research questions than ours.

Our study has several advantages, such as a large population-based sample with a high participation rate. Nevertheless, the findings must be interpreted in the light of some limitations: First, the latent genetic factor includes variance shared by the polygenic score and the mated phenotype. If the phenotypes are differentially operationalized or assessed in discovery and target samples, h and s may be smaller and the residuals larger. Thus, the interpretation of the model depends on the similarity between the GWASed and target phenotype and is only straightforward when these are the same. Second, the rGenSi model makes certain simplifying assumptions. Extensive simulations indicated that the model provided the expected results. In addition, the empirical results that could be compared with previous studies were mainly in line with these. However, we did not include covariance between an individual's latent genetic and environmental factors, which would be expected under vertical transmission. Assuming independence is a limitation of the current model and it remains to be determined how this influences the results in various scenarios. We also relied on previous GWAS to construct the polygenic scores. Assortative mating and parental indirect genetic effects may have influenced these studies and thereby also inflated our estimates of heritability. If we have overestimated heritability, genetic correlations between relatives will most likely also be inflated. Recent studies are exploring how these phenomena influence genomic studies and how to counteract this[43–45]. Third, all participants were expecting parents as our sample was based on a pregnancy cohort, and we could not study the role of selective fertility for genetic variance. Fourth, there is some selective participation in MoBa[46] that is plausibly enhanced among families where both parents participate. Fifth, although we attempted to adjust for population structure by including the first 50 principal components, we cannot rule out the effects of residual population structure[28,47]. For example, some parts of the spousal correlations could be due to assortment on social factors such as place of birth rather than assortment on phenotypes. Sixth, the MoBa sample is homogenous and mostly of European ancestry, potentially constraining generalizability to similar ethnicities.

This paper has four key findings. First, correlations between relatives' polygenic scores may reflect only a part of the genetic similarity arising from assortative mating. We have presented a structural equation modelling approach to estimate genetic correlations between individuals when phenotypic data is available. Second, the application of this model to partners, siblings, and in-laws indicated high levels of genetic partner similarity for educational attainment and some partner similarity for height and depression. Phenotypes were correlated with partners' and, in some cases, in-laws' polygenic scores. Third, partner similarity in

educational attainment and, to a smaller degree, in depression appeared not to be based directly on these phenotypes but rather on correlated phenotypes that we did not observe. Partner similarity in height resulted from direct phenotypic assortment only. Fourth, the genetic variances of educational attainment, height, and depression appeared to be in intergenerational equilibrium. Even though higher education has only been widespread for a few generations, individuals may in previous times have assorted on traits that predict educational attainment today. Hence, this study shows genetic similarities between extended family members and that assortative mating has taken place for several generations.

## Methods

**Ethics**. The establishment of The Norwegian Mother, Father, and Child Cohort Study (MoBa) and initial data collection was based on a license from the Norwegian Data Protection Agency and approval from The Regional Committees for Medical and Health Research Ethics. The MoBa cohort is now based on regulations related to the Norwegian Health Registry Act. The current study was approved by The Regional Committees for Medical and Health Research Ethics, Southern and Eastern Norway (project# 2017/2205). Informed consent was obtained from all study participants. The consent allows linking with data from other sources. The participants did not receive monetary compensation.

**Sample**. MoBa is a population-based pregnancy cohort study conducted by the Norwegian Institute of Public Health[48]. Participants were recruited from all over Norway from 1999 to 2008. The women consented to participation in 41% of the pregnancies. The cohort now includes 114,500 children, 95,200 mothers and 75,200 fathers. The current study is based on version 12 of the quality-assured data files.

We use data on the parent generation in the MoBa sample, that is, mothers and fathers but not children, and study the following relationship types among them, detailed in Fig. 1: Partners are the opposite-sex genetic parents of a child, regardless of their past or current relationship status. Siblings descend from the same two parents in the generation before the one represented in our study sample. We do not use data from individuals with unknown parents. In-laws (siblings-in-law) are separated by two degrees, one of partnership and one of siblingship, that is, either one's sibling's partner or one's partner's sibling. Co-in-laws (co-siblings-in-law) are the respective partners of siblings and are thus separated by three degrees: two of marriage and one of siblingship, that is, one's partner's sibling's partner. We identified siblings and their partners mapped through a link with the population register at Statistics Norway. This way, we identified 63,781 complete pairs of partners, 13,455 complete pairs of siblings, 21,496 complete pairs of in-laws, and 8699 complete pairs of co-in-laws (after removing 3 extended families where the co-in-laws were siblings). Women were on average 30.62 (SD = 4.66) years old and men were on average 33.18 (SD = 5.37) years old.

Blood samples were obtained from both parents during pregnancy. After birth, a second blood sample was taken from the mother. Genotyping of the entire MoBa cohort is ongoing. We used genotype data based on 32,000 family trios. The analytic sample included genotype data on 30,197 mothers and 28,691 fathers. There were valid genomic data on 26,681 complete couples, 2170 complete sibling pairs, 3905 in-law pairs, and 1763 co-in-law pairs (after removing 4 extended families where the co-in-laws were siblings). Information about the genotyping, imputation, and quality control is available in the Supplementary Methods 1 and Supplementary Table 1 and in previous reports[49,50].

## Measures

*Educational attainment*. Educational attainment of the participants at age 30 was gathered from registers at Statistics Norway and coded according to the Norwegian Standard Classification of Education[51]. Seven categories were in use, corresponding to 1 = Lower secondary school (9 years; mandatory education), 2 = Upper secondary school, basic (10-11 years), 3 = Upper secondary school, completed (12 years), 4 = Post-secondary non-tertiary education, 5 = Bachelor's degree or equivalent, 6 = Master's degree or equivalent, 7 = PhD or equivalent. Level of education (1–7) was used as the unit on these variables.

*Height*. Height in centimeters was self-reported by mothers and fathers in the first questionnaire (15 weeks of gestation).

*Depression*. Symptoms of major depressive disorder were measured with 6 items constituting the Life-Time History of Major Depression, which is based on DSM-III[52]. The average of the items is used as an individual's score on depression. The genetic association between major depressive disorder and broader definitions of depression has been reported to be very high ($r = 0.86$)[53]. The internal consistency in this sample was Cronbach's $\alpha = 0.82$ for women and Cronbach's $\alpha = 0.81$ for men.

*Polygenic scores.* Polygenic scores were calculated using PRSice2, based on European samples from the most recent GWAS of educational attainment[54], height[55], and major depressive disorder[56]. The main results are based on SNPs associated with the phenotype at p<0.05 in the GWAS. The Supplementary Methods 1 provide further information on the polygenic scoring method and Supplementary Note 2, Supplementary Figs. 7 and 8 provide sensitivity analyses where the main analyses are re-run using nine different thresholds.

**Statistical modelling.** We used standardized residuals for all variables (mean = 0, SD = 1) adjusted for population structure (top 20 principal components) and batch effects for polygenic scores. We estimated Pearson correlations between relatives' phenotypes and polygenic scores and across phenotypes and polygenic scores. Including the principal components in the analyses only marginally changed the correlations (see Supplementary Table 8). We provide 95% confidence intervals for all estimates and perform null hypothesis significance tests with a 5% alpha level, with the null value set to either 0.00 or 0.50 depending on the parameter in question.

*The correlation in genetic signal (rGenSi) model of phenotypes and polygenic scores.* The correlation between two individuals' polygenic scores may not correspond to the full resemblance on causal variants for the trait because a polygenic score may not be identical to the weighted sum of true trait-associated genetic variants in the target sample. We introduce a structural equation model (SEM), illustrated in Fig. 2, where the genetic predispositions are included as a latent variable. This latent variable influences both the phenotype and the polygenic score. It reflects variance shared by these and is usually not identical to any of them. The correlation between an individual's phenotype and polygenic score can then be divided into two components: i) the correlation between the latent genetic variable and the polygenic score, $s$, and ii) the correlation between the latent genetic variable and the phenotype, $h$. Usually, the correlation between phenotype and the polygenic score (the product $h * s$) can be observed, but separating the two components is impossible. Using data from relatives, we can identify $h$ and $s$ and provide correlations in the genetic signal free from the residual noise. We call this model rGenSi because we use it to study correlations in genetic signals among relatives. The model estimates the genetic correlation ($\hat{r}_g = r_{latent\ genetic}$) between individuals using data on polygenic scores and the corresponding phenotypes. The model also provides estimates of the proportion of genetic signal in the polygenic score ($s^2$), the proportion of noise ($n^2 = 1-s^2$), and the heritability ($h^2$) of a phenotype, and it can be extended to distinguish between direct and indirect assortment.

The association between partners' phenotypes is modelled as a co-path, $m$, that results from direct (primary phenotypic) assortment. The correlation between siblings' genetic signals is estimated in the model as $\widehat{r_{g,sibling}} = r_s$. Correlations between different individuals' genetic signals can be estimated by following path tracing rules allowing for co-paths. Co-paths connect valid chains of paths[20,32]. They contribute to the co-variance between variables, but not their variance. They are useful for analyzing assortment processes, which do not change the values of variables. For instance, the correlation between partners' genetic signals is $\widehat{r_{g,partner}} = h * m * h = mh^2$ and the correlation between co-in-laws' genetic signals is $\widehat{r_{g,co-in-law}} = h * m * h * r_s * h * m * h = r_s(mh^2)^2$.

The basic version of the rGenSi model has four free parameters, $s$, $h$, $m$, and $r_s$. It can be extended with two additional parameters, $a$ and $c$, which are fixed to 1 and 0, respectively, in the basic model. The association between the observed phenotype and a latent phenotype on which assortment takes place is represented by $a$. When $a = 1$, assortment takes place directly on the observed phenotype (primary phenotypic assortment). When it is lower, the assortment is indirect, as a correlated phenotype underlies the assortment (secondary assortment). For instance, academic abilities rather than obtained length of education could lead to partner similarity in education, and partner similarity in manifest depression might be due to similarity in neuroticism. The phenotype on which partners assort is modelled as a latent variable, like in the *Cascade* model[22]. When $a$ is freely estimated, $m$ corresponds to the correlation between a polygenic score and the partner's phenotype divided by the correlation with one's own phenotype. (A similar comparison underlies Robinson et al.'s[6] phenotypic estimation.) The $a$ parameter reduces all correlations between observed phenotypes to the same degree, that is, partner and co-in-law correlations are equally reduced by $a^2$.

Finally, siblings' phenotypes may be more similar than indicated by their genetic similarity, which means that they may be shaped by having similar environments. This can be modelled with a factor that is shared by siblings and that influences their phenotypes ($c$). The genetic residual is defined as not being associated with the phenotype. It is therefore assumed not to be a basis for partner selection and not to correlate between partners. Likewise, we assume that assortment in the previous generation is unrelated to the genetic residual, allowing us to set the residual genetic correlation to the uninflated value of 0.50 in siblings. The model is further explained in the Supplementary Methods 2, Supplementary Figs. 1–6, and Supplementary Tables 2–4, where we provide simulations showing that the model can simultaneously estimate all the parameters described here when the sample size is adequate. We used OpenMx 2.19.1 in R 4.0.2 to fit the models to raw data, providing full-information maximum likelihood estimates of all parameters.

*Testing intergenerational genetic equilibrium.* Assortative mating can lead to genetic correlations between partners and elevated genetic correlations between siblings in succeeding generations until an equilibrium is reached. In the rGenSi model, the genetic correlations between partners and between siblings are estimated independently. In equilibrium, however, the genetic correlation between partners can easily be predicted from the genetic correlation between siblings and vice versa. If the correlation between partners in additive genetic factors is $r_{g,partner}$, the additive genetic correlation between siblings in an equilibrium population is expected to be $\widehat{r_{g,sibling}} = \frac{1}{2}(1 + r_{g,partner})$[19] (p. 158). (The source states this as $r_{g,sibling} = \frac{1}{2}(1+\rho_z h^2)$, but we define $\rho_z h^2 = r_{g,partner}$). This formula allows us to compare partner correlations, which result from the assortment in the current generation, with sibling correlations, which results from the assortment in previous generations. An observed partner correlation higher than expected in equilibrium indicates that the level of genetic assortment is increasing compared to previous generations. We investigated deviations from intergenerational equilibrium by constraining correlations between partners' and siblings' polygenic scores according to the above formula and by adding this as a constraint to the rGenSi model. We compared models that assumed equilibrium to the less restricted models that did not assume equilibrium and tested whether they had a significantly worse fit to the data.

**Reporting summary.** Further information on research design is available in the Nature Research Reporting Summary linked to this article.

## Data availability
The MoBa data are available under restricted access due to data privacy laws, access can be obtained by application to MoBa and a Regional Committee for Medical and Health Research Ethics in Norway. The data in this study were accessed under ethics approval (project# 2017/2205, Regional Committees for Medical and Health Research Ethics, Southern and Eastern Norway). If you would like to apply for access, please see the following website for more details https://www.fhi.no/en/studies/moba/for-forskere-artikler/research-and-data-access/.

## Code availability
Scripts for the rGenSi model and the simulations are provided in Supplementary Software 1 and at https://osf.io/v9ybu/.

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

## Acknowledgements

This work was supported by the Research Council of Norway (#300668 to F.A.T., and #262177 to E.Y.). This work was partly supported by the Research Council of Norway through its Centres of Excellence funding scheme (#262700, #223273). A.H. and L.J. Hannigan were supported by the South-Eastern Norway Regional Health Authority (#2020022 and #2018058 to A.H.). R.C., A.H., and E.Y. were supported by the Research Council of Norway (#288083). The Norwegian Mother, Father and Child Cohort Study is supported by the Norwegian Ministry of Health and Care Services and the Ministry of Education and Research. We are grateful to all the participating families in Norway who take part in this ongoing cohort study. The genotyping, quality control, and imputation were supported by Norwegian Institute of Public Health, the HARVEST collaboration, NORMENT Centre, the Research Council of Norway (#229624, #223273, #240413), KG Jebsen Stiftelsen, the European Research Council (#293574), Stiftelsen Kristian Gerhard Jebsen, Trond Mohn Foundation, the Novo Nordisk Foundation (#54741), the Center for Diabetes Research at the University of Bergen, and the Western Norway Regional Health Authorities (Helse Vest).

## Author contributions

F.A.T. conceived of the idea and designed the model, whereas E.M.E., E.Y., R.C., and L.J. Hannigan contributed to the design and implementation of the research. R.C. and L.J. Hannigan contributed to sample preparation and quality control of genomic data and polygenic scores with support from A.H. F.A.T. carried out the analyses with support from E.M.E. F.A.T. wrote the manuscript with input from all authors. R.C. wrote parts of the supplement. F.A.T. planned and carried out the simulations with help from E.M.E. L.J. Howe contributed to the interpretation of the results. P.M., T.R.-K., O.A.A., and P.N. contributed to data generation and acquisition. A.H. and E.Y. supervised the project. All authors provided critical feedback, discussed the results and helped shape the manuscript.

## Competing interests

The authors declare no competing interests.
