## [Peer Review File · Nature Communications]

Modeling assortative mating and genetic similarities between partners, siblings, and in-lawsReviewers' Comments:

Reviewer #1:

Remarks to the Author:

In this study, Torvik et al. propose a new model to detect assortative mating (AM) by contrasting close relatives, spouses and in-laws and apply this model to show evidence of AM on height, educational attainment (EA) and susceptibility to depression. The authors claim that correlation of genetic predictors of traits lead to underestimate the strength of AM and that AM on EA may have not yet reached equilibrium in the Norwegian population. Overall, I found the paper well written but have identified major limitations to the present study, which limit the reliability and strength of the conclusions.

Major comments

1/ Correlation of polygenic scores (PGS) is biased. I don't think this statement is correct. Robinson et al (2017) [ref. 11] showed that the correlation of PGS accurately reflects the strength of assortment only when the PGS has the "BLUP property", i.e., when $E[\text{trait}|\text{PGS}] = \text{PGS}$. As this property is not systematic for all PGS, the statement made by the authors is therefore incorrect, unless proven otherwise. Yengo et al. (2018) [ref. 10] derived the expected of the correlation between PGS of spouses (under simplifying assumptions) as 2θ where θ was given in the Eq. (1) of that study. Can the authors show that their estimate is not expected under this simple theoretical framework.

Another observation, which may contribute to explain the (supposedly correlation between PGS) is that the authors have in fact analysed PGS after regressing out principal components (PCs). This analysis is not equivalent to fitting the following model: $\text{PGS}_{\text{spouse1}} = \text{PGS}_{\text{spouses2}} + \text{PCs}$. In fact, the approach used by the authors can actually induce a negative correlation between PGS of spouses. The magnitude of this bias depends on how much PC are correlated with PGS.

Also, the authors seem to have selected the best PGS that correlates with the trait of interest in the same sample used to infer AM. I expect such procedure to induce biases, in particular when using sub-significant SNPs as done in this study.

2/ Assortative mating on depression. I found this result to be most interesting as up until now limited evidence has been provided about AM on psychiatric traits. However, it seems like this result is not robust enough. First, although no p-value was provided for it we can infer a p-value just under 0.05 given the width of the 95% confidence interval. The level of significance is not robust enough also owing to all potential issues with PGS analyses listed above.

Minor comments

The authors haven't numbered the lines, which makes it difficult to refer to.

Sentence: "Increased genetic variation is followed by more individuals with high and low phenotypic scores" is confusing to me.

Sentence: "detectable via imprinting on individuals' genomes". This is confusion as "imprinting" as specific meaning in genetics. Also, the signature detected in that paper is at the population level not that of specific individuals.

"Choosing a higher alpha threshold did not substantially change partner correlations in polygenic scores, whereas choosing a lower alpha threshold reduced them somewhat, as detailed in Supplement B". This is not quantitative enough. What exactly is meant by "did not change substantially"?

Did the author check the genetic relatedness between in-laws and co-in laws. It was not mentioned

(apologies if I missed it) whether genetic relatedness was assessed empirically (i.e. not just relying on pedigree information)

The authors should report estimates of heritability obtained with their methods so that the reader can appreciate consistency with previously published data.

“correlation between exactly two individuals” That sounds incorrect. The correlation is calculated for many pairs of individuals, not just two people.

The expected relationship between fullsib under AM was established way before ref. 18. For example Crow and Felsenstein 1968.

Reviewer #2:

Remarks to the Author:

Summary

The authors used structural equation modeling (SEM) to gain insights into phenotypic and genetic similarity between different types of relatives in three traits. The authors created a correlated factors model (rGenSi) that allows them to estimate the latent (full) genetic correlations (r_g) between the four types of relatives—in-laws, co-in-laws, siblings, and partners. In doing so, the authors are able to detect genetic signatures of assortative mating (AM; e.g., the r_g between mates or in-laws, or the increased $r_g > .5$ for siblings). Furthermore, by comparing the r_g between mates and that expected between siblings for that r_{g_mate} , the authors demonstrate that AM for educational attainment is not at equilibrium.

There is a kernel of a good idea here, but the presentation is a bit confusing and more needs to be done to flesh out the interpretation of the results as well as to ensure that the results are correct (which I get to below). There are two innovations here. Neither is completely novel—rather they build on previous approaches—but in my opinion they could justify publication in Nature Communication:

- Use of SEM to correct for error in the PGSs, which enables estimation of the unbiased r_g between relative types. This achieves the same goals (using a different approach) as the Robinson (2017) paper that was cited, which instead extrapolates the r_g from PGS's built from BLUP estimates. A similar approach of using PGS's within SEM to test models of AM was also presented in a recent paper by Balbona et al. (2021).
- Estimation of r_g between different relative types in order to come to conclusions about whether AM for 3 traits is under equilibrium. Kong et al had come to the same conclusion about AM for educational attainment not being at equilibrium. They used a conceptually similar but nevertheless slightly different approach (finding that the correlation of PGS's across mates was higher than the correlation of PGS's within individuals). So it is nice that this paper, using a different method and different dataset, comes to the same conclusion. Balbona et al. also described how to use SEM to test for equilibrium vs. disequilibrium AM.

It would help the paper for the authors to be more clear about what is novel and to place it in better context in the literature. The authors are indeed doing something slightly different than any of the papers mentioned above, but it is similar enough that these papers should be cited and discussed and the approaches compared. Only the Robinson paper is mentioned, and I believe their description of the Robinson approach is incorrect (see below).

Below, I discuss specific issues that could be used to improve the paper in the order in which they appear.

- 1) p. 3 – there are many possible causes of mate similarity – clarify the authors are talking about primary phenotypic assortment

2) p. 3 – authors state that educational attainment is evolutionarily novel and “the genetic variation may therefore not yet have stabilized.” This is a bit sloppy. AM leads to equilibrium within about 5-10 generations, so the evolutionary timescale is irrelevant here.

3) P. 3 – “imprinting on individual’s genomes”. I know this was in the title of Yengo’s paper, but the authors should be clearer about what AM is doing in the genome (e.g., creating gametic phase disequilibrium).

4) P. 4 – when first introduced, the “in-law” relative type needs to be defined. There are many types of in-laws (mother-in-law, son-in-law, uncle-in-law, etc.). The authors are talking about one specific type: sibling-in-law. And similarly for “co-in-law”.

5) P. 4 – I found their description of polygenic scores to be vague and unhelpful. I would rework this, something along the lines that they predict the trait a) to the degree the trait is heritable; b) to the degree that measured SNPs are in LD with causal variants (mostly common); and c) to the degree that estimated effects correspond to the true association.

6) P.4 – bottom – as noted above, their description of why the PGS for EA goes from $r=.18$ to $r=.65$ is a misunderstanding of what is shown in Robinson Fig 2. BLUP has a minor impact on increasing the correlation of PGS between partners or between the PGS and the trait (e.g., from $r=.11$ to $r=.12$, see Robinson supp figure 2). If it were the case that BLUP recovered the $\text{SNP}-h^2$, everyone would be using BLUP to create PGS’s! But BLUP has the nice property of having a slope of 1 when $Y \sim \text{PGS_BLUP}$ vs. $Y \sim \text{PGS_OLS}$. That property allows one to see what the predicted partner correlation is, as extrapolated from the $Y_{\text{male}} \sim \text{PGS_BLUP}_{\text{female}}$ and vice-versa (this occurs simply by standardizing the PGS_BLUP and Y variables). It is those predicted correlations that are shown in Fig 2, and these predictions, based on the PGS_BLUP, that are shown in Figure 2.

7) P. 7 – what type of reliability alpha is being presented? Cronbach?

8) P. 8 – why do the authors residualize the PCs and other covariates out of the PGS? This can lead to biases vs. accounting for covariates properly in the same model (or by regressing the covariates out of both the PGS and the phenotype). Doing this properly should be easy to do in the means part of an openmx model.

9) P. 8 – details needed on how the PCs were calculated (e.g., within-sample only or anchoring to a reference panel?). Have authors done a sensitivity analysis controlling for more (e.g., 20 or 50) PCs? This isn’t a minor issue here, as stratification is the most obvious competing explanation for the results the authors present.

10) General – the authors have a new model. The standard in genetics is to test that the model works as intended by simulating genetic data and ensuring that your estimates agree with the parameters you put into the simulation. I strongly urge the authors to do this. Simulating genetic data under AM may be tricky but there are ways to do this (e.g., GeneEvolve).

11) P. 9 – strange citation for the expected rg_{sibling} . Perhaps a standard text, such as Lynch & Walsh? Or Falconer & Mackay?

12) P. 9 as well as Figure 4 and results therein – authors are merely testing whether rg_{sib} is higher than what’s expected based on rg_{partner} . I agree that tests what the author says it does, but it doesn’t go far enough to elucidate the degree to which the trait is not at equilibrium. Preferably, the authors could provide quantitative guidance about what kinds of discrepancies would be expected under different types of disequilibrium (e.g., 0, 1, 2, 3, 4, 5 ...inf generations of AM). There may be

closed-formed ways of doing this, but if not, again, simulation might help.

13) The PGS scores already have the influences of genetic nurture and AM embedded within them. What influence does this have on the authors' results?

14) P. 11 – Checking intergenerational equilibrium...section. What exactly are the predicted correlations based on? This isn't clear enough. The predicted $r_{g_partner}$ is based on the observed sib correlation under the assumption of equilibrium? Whereas the predicted r_{g_sib} is based on the observed $r_{g_partner}$ under the equilibrium assumption? Or what exactly? This is central to the paper but not well enough explained.

15) P.11 – I suggest simplifying the results/presentation by just focusing on the extended rGenSi model (which are what are presented in the figures anyway) and forgetting about the simpler rGenSi individual results.

16) P. 12 – use of the in-law is interesting. In general, I feel like the authors have missed an opportunity to discuss these results in general as being a test of primary AM vs. other mechanisms. These results are consistent with primary.

17) P. 14, "The inclusion of in-laws also made it possible to compare with results found using quantitative genetic methods, based on siblings and their partners [20, 31]. We found that the results were similar, despite that these models do not share assumptions with the molecular genetic models. They thus provided triangulation, increasing our confidence in the results." Where are these comparisons presented?

18) Discussion - I think there is a missed opportunity to expound on the most obvious reason for why education wouldn't be in equilibrium while the other traits are.

19) P. 21 – table 1 – the "Phenotype x Polygenic score" nomenclature is confusing. It's not clear what this means. I was thinking interaction when I saw it, but it's $r(Y, PGS)$.

20) P. 23 – fig 2 – seems too simple to put in its own figure. It's already in fig 3 anyway.

21) P. 24 – fig 3 – the residual terms shouldn't all use the same letter ("R")

22) P. 25 – fig 4 – the lines between the observed r and the $r_{genetic}$ signals is misleading (as if it were increasing), but the two are measuring different things.

23) P. 25 – fig 4 – needs to be explicit that this is from the extended rGenSi model (if the authors continue to use both models in the MS).

REVIEWER COMMENTS

Reviewer #1 (Remarks to the Author):

In this study, Torvik et al. propose a new model to detect assortative mating (AM) by contrasting close relatives, spouses and in-laws and apply this model to show evidence of AM on height, educational attainment (EA) and susceptibility to depression. The authors claim that correlation of genetic predictors of traits lead to underestimate the strength of AM and that AM on EA may have not yet reached equilibrium in the Norwegian population. Overall, I found the paper well written but have identified major limitations to the present study, which limit the reliability and strength of the conclusions.

RESPONSE: We thank the reviewer for a thorough evaluation and for constructive comments.

Major comments

1/ Correlation of polygenic scores (PGS) is biased. I don't think this statement is correct. Robinson et al (2017) [ref. 11] showed that the correlation of PGS accurately reflects the strength of assortment only when the PGS has the "BLUP property", i.e., when $E[\text{trait} | \text{PGS}] = \text{PGS}$. As this property is not systematic for all PGS, the statement made by the authors is therefore incorrect, unless proven otherwise. Yengo et al. (2018) [ref. 10] derived the expected of the correlation between PGS of spouses (under simplifying assumptions) as 2θ where θ was given in the Eq. (1) of that study. Can the authors show that their estimate is not expected under this simple theoretical framework.

RESPONSE: We assume that the reviewer refers to the second last paragraph in the introduction, where we wrote "polygenic scores are not perfect indicators of these predispositions". We do not claim that correlations between polygenic scores are always biased, but that they can be biased (just like the correlations between any measures). We have rewritten this entire paragraph (see below) to make it clear in which situations one can expect the correlations between polygenic scores to be underestimates of the true genetic correlation, and when they are not. To clarify, we do not claim that there is anything wrong with Yengo et al.'s framework, nor that similar results could not be obtained by using it on our data. We do, however, believe that structural equation modelling can address the questions we aimed to answer in this paper.

From manuscript page 5 line 1-17:

One way to investigate genetic resemblance between individuals is to calculate correlations between their polygenic scores. A polygenic score summarises an individual's genetic predisposition to a trait across many single nucleotide polymorphisms (SNPs) [23, 24]. Polygenic scores correlate with a trait to the degree that the trait is heritable and that the polygenic score captures that heritable component. Polygenic scores do this by including SNPs in linkage disequilibrium with causal variants weighted according to the true associations with the trait in the target sample. The variance in a trait explained by polygenic scores is typically lower than its heritability, indicating that polygenic scores usually do not fully capture the heritable component [25]. If polygenic scores are imperfectly correlated with the genetic predispositions to a trait, the correlation between multiple individuals' polygenic scores can be lower than the correlations between the true set of trait-associated genetic variants. For instance, correlations between spouses' polygenic scores could be underestimated if one partner has many trait-increasing alleles included in the polygenic score whereas the other has many trait-increasing alleles not included in the polygenic score. Polygenic scores for educational attainment have been found to correlate lower in partners, at only $r=0.18$ [26] or $r=0.11$ [27], than adjusted estimates of genetic correlations based on genetic predictors and the phenotypes of

partners ($r=0.65$) [9]. Several approaches to related scaling issues have been proposed [28, 29], including structural equation modelling [30].

Another observation, which may contribute to explain the (supposedly correlation between PGS) is that the authors have in fact analysed PGS after regressing out principal components (PCs). This analysis is not equivalent to fitting the following model: $PGS_{spouse1} = PGS_{spouses2} + PCs$. In fact, the approach used by the authors can actually induce a negative correlation between PGS of spouses. The magnitude of this bias depends on how much PC are correlated with PGS.

RESPONSE: To address the reviewer's comment, we examined correlations between individuals' polygenic scores and phenotypes a) with no adjustment for PCs, b) after residualizing on PCs, c) when including the PCs in the estimation of the correlations. The three different approaches have negligible differences in our data, as shown in Supplementary Table 7. We therefore conclude that residualizing on the PCs matters little in the current situation. See also our response to reviewer #2's comment #8.

Also, the authors seem to have selected the best PGS that correlates with the trait of interest in the same sample used to infer AM. I expect such procedure to induce biases, in particular when using sub-significant SNPs as done in this study.

RESPONSE: We did not select the polygenic scores that correlated most highly with the corresponding phenotype. In Supplementary Table 6 we compare correlations between spouses using different versions of the polygenic scores. In the revised version of the manuscript, we additionally provide full results for nine different versions of each polygenic score (see Supplementary Figures 7 and 8). An advantage of using the *rGenSi* model is that although different versions of the polygenic scores captures different proportions of the true genetic variation, the model provides reasonably similar estimates of genetic correlations between relatives regardless of SNP selection p-value threshold. In general, and across the three phenotypes, higher p-value thresholds appear to indicate smaller confidence intervals in the results.

2/ Assortative mating on depression. I found this result to be most interesting as up until now limited evidence has been provided about AM on psychiatric traits. However, it seems like this result is not robust enough. First, although no p-value was provided for it we can infer a p-value just under 0.05 given the width of the 95% confidence interval. The level of significance is not robust enough also owing to all potential issues with PGS analyses listed above.

RESPONSE: We assume that the reviewer refers to the correlation between partners polygenic scores, which is not significant. However, the polygenic score for depression is related to the phenotype ($r=0.11$, 95%CI 0.10, 0.12), albeit not as strongly as for height and educational attainment. With this modest correlation between phenotype and polygenic score, a high correlation between two polygenic scores cannot be expected. Nevertheless, we found significant correlations between the polygenic scores of one partner and the other partner's level of depression ($r=0.03$, 95%CI 0.02, 0.04). Furthermore, we found genetic correlations between partners ($r=0.08$, 95%CI 0.03, 0.11) in the *rGenSi* model, which draws on additional data. Whereas the correlations are lower for depression than for the other phenotypes, the confidence intervals are of similar width. We therefore believe that the low correlations are robust.

Minor comments

[Comment 1:] The authors haven't numbered the lines, which makes it difficult to refer to.

RESPONSE: We apologise for this inconvenience. Line numbers have been added.

[Comment 2:] Sentence: “Increased genetic variation is followed by more individuals with high and low phenotypic scores” is confusing to me.

RESPONSE: We have changed this sentence on page 3 line 9-10:

Increased genetic variation is followed by a larger variation between individuals in phenotypic expression.

[Comment 3:] Sentence: “detectable via imprinting on individuals’ genomes”. This is confusion as “imprinting” as specific meaning in genetics. Also, the signature detected in that paper is at the population level not that of specific individuals.

RESPONSE: We agree that this wording was potentially confusing. The sentence on page 3 line 21-23 has been changed.

Assortative mating in previous generations can also be detected in samples of genomes from unrelated individuals by estimating covariance between trait-associated loci in distant parts of the genome [13].

[Comment 4:] “Choosing a higher alpha threshold did not substantially change partner correlations in polygenic scores, whereas choosing a lower alpha threshold reduced them somewhat, as detailed in Supplement B”. This is not quantitative enough. What exactly is mean by “did not change substantially”?

RESPONSE: This subjective evaluation is no longer used in the manuscript. We now provide much more thorough sensitivity analyses than in the initial submission. Supplementary Fig. 7 on page 15 in the supplement presents model parameters from *rGenSi* models using polygenic scores with different thresholds, whereas Supplementary Fig. 8 similarly presents genetic correlations between relatives derived from these models. Using higher cut-offs was associated with increases in the genetic signal for all phenotypes and smaller confidence intervals for the *rGenSi* estimated genetic correlations. Otherwise, the results remained similar across thresholds.

[Comment 5:] Did the author check the genetic relatedness between in-laws and co-in laws. It was not mentioned (apologies if I missed it) whether genetic relatedness was assessed empirically (i.e. not just relying on pedigree information)

RESPONSE: Individuals were defined as in-laws and co-in-laws based on pedigree information on siblings and marriages. The design accounts for siblings (first degree relatives), however, it is possible for more distant relatives to be included in the dataset either in the same extended family (as in-laws, co-in-laws, or even partners) or in another extended family (possibly inducing some dependence in the data). As a sensitivity analysis, we excluded all 2., 3., and 4., degree relatives from the sample and calculated the correlations between relatives’ polygenic scores and between relatives’ phenotypes. The results did not change substantially ($\Delta 0.01$ at most). Please see supplementary material page 13:

We identified 1,492 pairs of second, third, or fourth-degree relatives in the sample by using the KING (Kinship-based INference for Gwas) software (<https://www.kingrelatedness.com/>) with the `-ibs` command (identity by state). Siblings (first-degree relatives) were already a part of the design. As a sensitivity analysis, we randomly excluded one individual from each pair and recalculated the correlations between relatives’ polygenic scores and relatives’ phenotypes. This led to changes in correlations of at most 0.01 compared to Supplementary Table 7.

[Comment 6:] The authors should report estimates of heritability obtained with their methods so that the reader can appreciate consistency with previously published data.

RESPONSE: We now report heritability (h^2) in the main manuscript, please see Figure 3. Exact numbers from the different models are provided in Supplementary Table 5 (as the model estimates h , this needs to be squared to obtain heritability).

[Comment 7:] “correlation between exactly two individuals” That sounds incorrect. The correlation is calculated for many pairs of individuals, not just two people.

RESPONSE: We thank the reviewer for pointing this out. The phrasing is not used in the revision.

[Comment 8:] The expected relationship between fullsib under AM was established way before ref. 18. For example Crow and Felsenstein 1968.

RESPONSE: We have changed this to Lynch & Walsh (1998), p. 158, cf. comment 11 by reviewer #2.

Reviewer #2 (Remarks to the Author):

Summary

The authors used structural equation modeling (SEM) to gain insights into phenotypic and genetic similarity between different types of relatives in three traits. The authors created a correlated factors model (rGenSi) that allows them to estimate the latent (full) genetic correlations (r_g) between the four types of relatives—in-laws, co-in-laws, siblings, and partners. In doing so, the authors are able to detect genetic signatures of assortative mating (AM; e.g., the r_g between mates or in-laws, or the increased $r_g > .5$ for siblings). Furthermore, by comparing the r_g between mates and that expected between siblings for that r_{g_mate} , the authors demonstrate that AM for educational attainment is not at equilibrium.

There is a kernel of a good idea here, but the presentation is a bit confusing and more needs to be done to flesh out the interpretation of the results as well as to ensure that the results are correct (which I get to below). There are two innovations here. Neither is completely novel—rather they build on previous approaches—but in my opinion they could justify publication in Nature Communication:

a) Use of SEM to correct for error in the PGSs, which enables estimation of the unbiased r_g between relative types. This achieves the same goals (using a different approach) as the Robinson (2017) paper that was cited, which instead extrapolates the r_g from PGS's built from BLUP estimates. A similar approach of using PGS's within SEM to test models of AM was also presented in a recent paper by Balbona et al. (2021).

b) Estimation of r_g between different relative types in order to come to conclusions about whether AM for 3 traits is under equilibrium. Kong et al had come to the same conclusion about AM for educational attainment not being at equilibrium. They used a conceptually similar but nevertheless slightly different approach (finding that the correlation of PGS's across mates was higher than the correlation of PGS's within individuals). So it is nice that this paper, using a different method and different dataset, comes to the same conclusion. Balbona et al. also described how to use SEM to test for equilibrium vs. disequilibrium AM.

It would help the paper for the authors to be more clear about what is novel and to place it in better context in the literature. The authors are indeed doing something slightly different than any

of the papers mentioned above, but it is similar enough that these papers should be cited and discussed and the approaches compared. Only the Robinson paper is mentioned, and I believe their description of the Robinson approach is incorrect (see below).

Below, I discuss specific issues that could be used to improve the paper in the order in which they appear.

RESPONSE: We thank the reviewer for a thorough discussion of our paper and for constructive comments. We also thank the reviewer for making us aware of Balbona et al.'s (2021) very relevant paper, which we now cite in the manuscript. We have reworked the presentation of the results and have made substantial changes in the introduction, discussion, and methods sections, and we have made new figures and conducted extensive simulations. In line with the reviewer's suggestion, we now devote much more attention to the process underlying partner similarity and model direct versus indirect assortment. This has led us to revise our model. Our simulations indicated that the previous version of the model was correct in some, but not all scenarios. A key feature of the revised model is the ability to discern direct versus indirect assortment. We now provide simulations showing that this revised model can accurately estimate the relevant parameters. We respond in more detail below.

1) p. 3 – there are many possible causes of mate similarity – clarify the authors are talking about primary phenotypic assortment

RESPONSE: An essential feature of the revised model is that we now analyse direct (primary) phenotypic assortment versus indirect (secondary) assortment. We have updated the introduction, the results, the discussion, methods sections, and the Supplementary Methods.

From introduction page 4, line 9-21:

In-laws and co-in-laws are only indirectly related, and data on such relatives are therefore informative for understanding the mechanisms leading to the resemblance between the partners that connect them. With direct assortment (also called primary phenotypic assortment), the similarity between partners results from assortment based on the phenotype in question. This can be distinguished from indirect assortment (also called secondary assortment), where partners resemble each other because they assort on one or more traits related to the trait of primary interest [19-21]. An example of direct assortment would be partner selection based on observed height. An example of indirect assortment would be partner similarity in education resulting from assortment on a proxy such as cognitive abilities. Under direct assortment, correlations between in-laws or co-in-laws should correspond to the product of the relations that connect them. Under indirect assortment, all correlations between relatives should be equally deflated. Data from in-laws and co-in-laws can therefore be used to separate between direct and indirect assortment.

From results page 7, line 5-14:

Figure 2 shows the full structural equation model. The parameter a is an estimate of the association between the observed phenotype and the latent phenotype that is the basis of partner selection. If $a=1$, there is only direct assortment, but if it is less, indirect assortment also takes place. The results of fitting the rGenSi model to the data and the estimated parameters of the different versions of the models are presented in Supplementary Tables 4 and 5. Figure 3 shows the parameter estimates for the rGenSi models with the best fit for each phenotype. Partner similarity in educational attainment ($a=0.77$, 95%CI 0.75, 0.78) and depression ($a=0.80$, 95%CI 0.69, 0.91) appeared to result from mating on proxy phenotypes highly related to, but not identical to these. Partner similarity in

height resulted from assortment directly based on observed height (a estimated at 0.97, 95% CI 0.92, 1.02, fixed to $a=1$ in the best fitting model).

From discussion page 9 line 21 – page 10 line 15:

Our models indicated that partner similarity in educational attainment and depression was partly due to indirect assortment on proxy characteristics, whereas partner similarity in height resulted from partner selected directly on manifest height. This means that “tall people choose other tall people because they are tall” [32](p. 382) and that there is nothing more to partner similarity in height. The direct assortment on height is in line with previous genetic studies [9]. For educational attainment and depression, the story is more complex. The association between observed education and the latent mated phenotype was not perfect, indicating that other traits than the observed played a role. The results for educational attainment are in line with previous studies finding that partner similarity in education results from indirect assortment on correlated traits [9, 33]. They are also consistent with reports of social homogamy in twin samples [34] because social homogamy and indirect assortment are indistinguishable in phenotypic data [19]. Individuals with highly educated families are likely to find highly educated partners, even if they do not have high education themselves, because unobserved proxy characteristics are responsible for the partner similarity. Our results cannot tell us what these characteristics are, but plausible candidates for educational attainment include cognitive abilities, conscientiousness, other indicators of academic abilities [35, 36], and the obtained education itself; most likely a combination of these. This could be clarified by investigating assortment on these traits in tandem with educational attainment. Our depression measure was intended to assess lifetime history of depression but might more strongly capture recent than past episodes [37]. It therefore seems plausible that factors underlying the observed partner similarity in depression could include depressive symptoms at the time of couple formation rather than the time of observation, neuroticism, or a general risk of psychopathology [38, 39].

From methods page 17, line 10-21:

The association between the observed phenotype and a latent phenotype on which assortment takes place is represented by a . When $a = 1$, assortment takes place directly on the observed phenotype (primary phenotypic assortment). When it is lower, the assortment is indirect, as a related phenotype underlies the assortment (secondary assortment). For instance, academic abilities rather than obtained length of education could lead to partner similarity in education, and partner similarity in manifest depression might be due to similarity in neuroticism. The phenotype on which partners assort is modelled as a latent variable, like in the Cascade model [21]. When a is freely estimated, m corresponds to the correlation between a polygenic score and the partner’s phenotype divided by the correlation with one’s own phenotype. (A similar comparison underlies Robinson et al.’s [9] phenotypic estimation.) The a parameter reduces all correlations between observed phenotypes to the same degree, that is, partner and co-in-law correlations are equally reduced by a^2 .

The topic is also addressed in the supplementary methods.

2) p. 3 – authors state that educational attainment is evolutionarily novel and “the genetic variation may therefore not yet have stabilized.” This is a bit sloppy. AM leads to equilibrium within about 5-10 generations, so the evolutionary timescale is irrelevant here.

RESPONSE: The relevant text on page 3 line 14 has been changed to “Furthermore, educational attainment has increased massively over the last few generations [16]. Therefore, the genetic consequences may not yet have fully unfolded.” We also show in Supplementary Figure 6 how the genetic variance is expected to increase after 0-15 generations with constant assortment.

3) P. 3 – “imprinting on individual’s genomes”. I know this was in the title of Yengo’s paper, but the authors should be clearer about what AM is doing in the genome (e.g., creating gametic phase disequilibrium).

RESPONSE: Reviewer #1 had the same comment (minor comment #3). We agree that this wording was potentially confusing. The sentence on page 3 line 21-23 has been changed.

Assortative mating in previous generations can also be detected in samples of genomes from unrelated individuals by estimating covariance between trait-associated loci in distant parts of the genome [13].

4) P. 4 – when first introduced, the “in-law” relative type needs to be defined. There are many types of in-laws (mother-in-law, son-in-law, uncle-in-law, etc.). The authors are talking about one specific type: sibling-in-law. And similarly for “co-in-law”.

RESPONSE: We now defined the terms and specify that we use the terms in-laws and co-in-laws for brevity. Please see page 4, line 6-9:

Resemblance between partners should induce resemblance, both phenotypic and genetic, between siblings-in-law (siblings of partners or partners of siblings) and by extension between siblings-co-in-law, who are the respective partners of siblings. For brevity, we refer to these relationship types as in-laws and co-in-laws, respectively.

We have also clarified this in the sample section on page 14 line 9-13.

In-laws (siblings-in-law) are separated by two degrees, one of partnership and one of sibblingship, that is, either one’s sibling’s partner or one’s partner’s sibling. Co-in-laws (siblings-co-in-law) are the respective partners of siblings and are thus separated by three degrees: two of marriage and one of sibblingship, that is, one’s partner’s sibling’s partner.

5) P. 4 – I found their description of polygenic scores to be vague and unhelpful. I would rework this, something along the lines that they predict the trait a) to the degree the trait is heritable; b) to the degree that measured SNPs are in LD with causal variants (mostly common); and c) to the degree that estimated effects correspond to the true association.

RESPONSE: We thank the reviewer for the suggestions on how to make this more clear and helpful. We have reworked the paragraph on page 5 line 1-17.

One way to investigate genetic resemblance between individuals is to calculate correlations between their polygenic scores. A polygenic score summarises an individual’s genetic predisposition to a trait across many single nucleotide polymorphisms (SNPs) [23, 24]. Polygenic scores correlate with a trait to the degree that the trait is heritable and the polygenic scores capture that heritable component. Polygenic scores do this by including SNPs in linkage disequilibrium with causal variants weighted according to the true associations with the trait in the target sample. The variance in a trait explained by polygenic scores is typically lower than its heritability, indicating that polygenic scores usually do not fully capture the heritable component [25]. If polygenic scores are imperfectly correlated with the genetic predispositions to a trait, the correlation between multiple individuals’ polygenic scores can be lower than the correlations between the true set of trait-associated genetic variants. For instance, correlations between spouses’ polygenic scores could be underestimated if one partner has many trait-increasing alleles included in the polygenic score whereas the other has many trait-increasing alleles not included in the polygenic score. Polygenic scores for educational attainment have been found to correlate lower in partners, at only $r=0.18$ [26] or $r=0.11$ [27], than

adjusted estimates of genetic correlations based on genetic predictors and the phenotypes of partners ($r=0.65$) [9]. Several approaches to related scaling issues have been proposed [28, 29], including structural equation modelling [30].

6) P.4 – bottom – as noted above, their description of why the PGS for EA goes from $r=.18$ to $r=.65$ is a misunderstanding of what is shown in Robinson Fig 2. BLUP has a minor impact on increasing the correlation of PGS between partners or between the PGS and the trait (e.g., from $r=.11$ to $r=.12$, see Robinson supp figure 2). If it were the case that BLUP recovered the SNP- h^2 , everyone would be using BLUP to create PGS's! But BLUP has the nice property of having a slope of 1 when $Y \sim \text{PGS_BLUP}$ vs. $Y \sim \text{PGS_OLS}$. That property allows one to see what the predicted partner correlation is, as extrapolated from the $Y_{\text{male}} \sim \text{PGS_BLUP}_{\text{female}}$ and vice-versa (this occurs simply by standardizing the PGS_BLUP and Y variables). It is those predicted correlations that are shown in Fig 2, and these predictions, based on the PGS_BLUP, that are shown in Figure 2.

RESPONSE: We thank the reviewer for pointing out this mistake. This has been rewritten in the introduction page 5 line 13-16.

Polygenic scores for educational attainment have been found to correlate lower in partners, at only $r=0.18$ [26] or $r=0.11$ [27], than adjusted estimates of genetic correlations based on genetic predictors and the phenotypes of partners ($r=0.65$) [9].

The similarity between Robinson's and our approach is also pointed out in the methods, page 17, line 19:

(A similar comparison underlies Robinson et al.'s [9] phenotypic estimation.)

7) P. 7 – what type of reliability alpha is being presented? Cronbach?

RESPONSE: Correct. This is now specified at page 15 line 17 and 18.

8) P. 8 – why do the authors residualize the PCs and other covariates out of the PGS? This can lead to biases vs. accounting for covariates properly in the same model (or by regressing the covariates out of both the PGS and the phenotype). Doing this properly should be easy to do in the means part of an openmx model.

RESPONSE: This approach leads to negligible differences in our data. The reviewer suggests adding the PCs to the means structure of the OpenMx models. We agree that that this is easily be specified in a script. However, the number of estimated parameters increase massively; there are 20 PCs for 4 individuals, each participating with 2 variables (phenotype and polygenic score). This implies at least $20 \times 2 = 40$ extra parameters, but approximately 4 times as many (160 extra parameters) when adjusting for relatives' PCs as well. We tried to do this, but found that the running time of the models became very long; the models were unstable and it was difficult to get them to converge. We have therefore adjusted for the principal components in pairwise estimation of correlations between the variables and show that this, in our case, leads to only negligible differences in observed correlations as compared to estimating the correlations after residualizing on the PCs. As the only show negligible differences, we expect that the SEM would also provide similar results. We therefore argue that residualizing on the PCs matters little in the current situation. In the Supplementary Table 7 we show correlations between individuals' polygenic scores and phenotypes a) with no adjustment for PCs, b) after residualizing on PCs, c) when including the PCs in the estimation of the correlations.

9) P. 8 – details needed on how the PCs were calculated (e.g., within-sample only or anchoring to a reference panel?). Have authors done a sensitivity analysis controlling for more (e.g., 20 or 50) PCs? This isn't a minor issue here, as stratification is the most obvious competing explanation for

the results the authors present.

RESPONSE: We have adjusted for 20 PCs, as the reviewer suggests (instead of 10), and provide more information in the supplement.

We have added the following information to the Supplementary Material on page 1 line 19-22:

Population stratification was assessed, using the HapMap phase 3 release 3 as a reference, by principal component analysis using EIGENSTRAT version 6.1.4. Visual inspection identified a homogenous population and individuals of non-European ancestries were removed based on principal component analysis of markers overlapping with available HapMap markers.

We have added the following information to the Supplementary Material on page 13 line 22-37:

To evaluate the influence of different methods for adjusting for principal components, we calculated associations between relatives' polygenic scores and relatives' phenotypes using three methods. These principal components are described in more detail in Supplement Methods, page 1. The three sets of correlations between relatives are shown in Supplementary Table 7. In the first set of results (a), we estimated crude associations between polygenic scores without taking the principal components into account. In the second set of results (b), we first residualised polygenic scores and phenotypes on 20 principal components and then estimated the associations between these residuals. In the third set of results (c), we included the principal components directly in the model used to estimate the correlations. The correlations were estimated in OpenMx, and the principal components were set to influence the means in the model. The results indicate that the three methods for estimating the correlations provide similar results, with differences in the second decimal place. Most differences were 0.00 or 0.01, with the largest being 0.03 (correlations between siblings' polygenic scores for height). We consider these differences to be trivial and of no practical importance. The small consequences of adjusting for the principal components may be due to the homogeneity among participants in the Norwegian Mother, Father and Child Cohort Study (MoBa), from which we draw our sample.

10) General – the authors have a new model. The standard in genetics is to test that the model works as intended by simulating genetic data and ensuring that your estimates agree with the parameters you put into the simulation. I strongly urge the authors to do this. Simulating genetic data under AM may be tricky but there are ways to do this (e.g., GeneEvolve).

RESPONSE: We now provide simulations in the Supplementary Methods. The simulations are based on a sample size of either 1,000 or 50,000, assortative mating over 10 generations. Two versions of the model have been run, each 1,000 times from random starting values for each of the sample size (4,000 simulations in total). The simulations show that the means, medians, and standard deviations produced by the rGenSi model come close to the true values. Please see Supplementary Figures 2, 4, and 5 and Supplementary Tables 2 and 3. In brief, the simulations indicate that the revised model can provide reasonable estimates of the true values.

11) P. 9 – strange citation for the expected $rg_{sibling}$. Perhaps a standard text, such as Lynch & Walsh? Or Falconer & Mackay?

RESPONSE: We agree that this looked strange. We now refer to Lynch & Walsh (1998) for the expected genetic correlation in equilibrium.

12) P. 9 as well as Figure 4 and results therein – authors are merely testing whether rg_{sib} is higher than what's expected based on $rg_{partner}$. I agree that tests what the author says it does, but it

doesn't go far enough to elucidate the degree to which the trait is not at equilibrium. Preferably, the authors could provide quantitative guidance about what kinds of discrepancies would be expected under different types of disequilibrium (e.g., 0, 1, 2, 3, 4, 5 ...inf generations of AM). There may be closed-formed ways of doing this, but if not, again, simulation might help.

RESPONSE: We agree that that would have been ideal, but as we no longer find significant deviations from equilibrium, we cannot quantify the distance from it (plausibly 0 generations away from equilibrium). We have added a Supplementary Figure 6 which shows that one gets values close to the expectation in equilibrium after ~5 generations with assortment. This is also discussed on page 10 line 17 – page 11 line 15, which is copied under our response to this reviewer's comment #1 (see above).

13) The PGS scores already have the influences of genetic nurture and AM embedded within them. What influence does this have on the authors' results?

RESPONSE: The polygenic scores are likely to be influenced both by genetic nurture and by assortative mating in previous generations. For instance, the polygenic score for educational attainment may include genetic variants related to parenting behaviours that increase educational attainment. In other environments, these influences may have been different. The polygenic scores were created in the same way for all study participants, regardless of whether they were included as partners, siblings, et cetera – most participants are included both as partners and as siblings at the same time. These issues should therefore have influenced the participants equally (if at all). Therefore, we do not believe that these issues would lead to systematic bias in our results.

14) P. 11 – Checking intergenerational equilibrium...section. What exactly are the predicted correlations based on? This isn't clear enough. The predicted $r_{g_partner}$ is based on the observed sib correlation under the assumption of equilibrium? Whereas the predicted r_{g_sib} is based on the observed $r_{g_partner}$ under the equilibrium assumption? Or what exactly? This is central to the paper but not well enough explained.

RESPONSE: We first estimate partner correlations and sibling correlations making no assumptions regarding equilibrium. We then assume equilibrium where these correlations are related to each other according to a specific formula. In the first model, the partner and sibling correlations are estimated independently, but not in the second. We then compare these two models and test which one that has the best fit to the data. We have updated the methods description on page 18 line 11-26:

Assortative mating can lead to genetic correlations between partners and elevated genetic correlations between siblings in succeeding generations until an equilibrium is reached. In the rGenSi model, the genetic correlations between partners and between siblings are estimated independently. In equilibrium, however, the genetic correlation between partners can easily be predicted from the genetic correlation between siblings and vice versa. If the correlation between partners in additive genetic factors is $r_{g,partner}$, the additive genetic correlation between siblings in an equilibrium population is expected to be $r_{g,sibling} = \frac{1}{2}(1 + r_{g,partner})$ [18] (p. 158). (The source states this as $r_{g,sibling} = \frac{1}{2}(1 + \rho_z h^2)$, but $\rho_z h^2 = r_{g,partner}$.) This formula allows us to compare partner correlations, which result from assortment in the current generation, with sibling correlations, which results from assortment in previous generations. An observed partner correlation higher than expected in equilibrium indicates that the level of genetic assortment is increasing compared to previous generations. We investigated deviations from intergenerational equilibrium by constraining correlations between partners' and siblings' polygenic scores according to the above formula and by adding this as a constraint to the rGenSi model. We compared models that assumed equilibrium to

the less restricted models that did not assume equilibrium and tested whether they had a significantly worse fit to the data.

15) P.11 – I suggest simplifying the results/presentation by just focusing on the extended rGenSi model (which are what are presented in the figures anyway) and forgetting about the simpler rGenSi individual results.

RESPONSE: We now focus exclusively on a revised version of the extended family model (a key feature of this model is the ability to distinguish between direct and indirect assortment).

16) P. 12 – use of the in-law is interesting. In general, I feel like the authors have missed an opportunity to discuss these results in general as being a test of primary AM vs. other mechanisms. These results are consistent with primary.

RESPONSE: We agree that use of in-laws provides interesting possibilities, such as testing direct vs indirect assortment. Testing direct (primary) phenotypic assortment versus indirect (secondary) assortment is now an important aspect of the paper and key feature of the revised model. We have copied relevant aspects of the paper under reviewer #2's comment #1 ("p. 3 – there are many possible causes of mate similarity – clarify the authors are talking about primary phenotypic assortment").

17) P. 14, "The inclusion of in-laws also made it possible to compare with results found using quantitative genetic methods, based on siblings and their partners [20, 31]. We found that the results were similar, despite that these models do not share assumptions with the molecular genetic models. They thus provided triangulation, increasing our confidence in the results." Where are these comparisons presented?

RESPONSE: This was previously presented in the supplement, with genetic correlations between partners also presented imposed on a figure in the main manuscript. However, we have removed the twin-and-sibling analyses from the revised paper (and the supplement) because they were based on mating only on the observed phenotype only (direct assortment) and therefore not comparable to the revised version of our rGenSi model. (The twin models also have to make assumptions regarding intergenerational equilibrium.)

18) Discussion - I think there is a missed opportunity to expound on the most obvious reason for why education wouldn't be in equilibrium while the other traits are.

RESPONSE: In the updated model, this finding has changed - we now only find this if we make the unrealistic assumption of direct assortment. The comment is therefore not relevant to the revised version of the manuscript. We have updated the discussion regarding equilibrium on page 10 line 17 – page 11 line 15.

A genetic partner correlation can lead to increased genetic variation in the next generation and a sibling correlation elevated above 0.50. This is exactly what we found for educational attainment, with a genetic sibling correlation of 0.68 (95% CI 0.61, 0.75). This indicates that the participants' parents were also engaged in assortative mating, a finding that is in line with previous findings of correlations between trait-associated alleles in different parts of the genome [13]. Higher education has been widely available in Norway for only a few generations, but people may have always selected partners based on proxy traits that are related to educational attainment today. Our results on direct versus indirect assortment indicate that such factors were important, meaning that assortment on genetic factors related to educational attainment is likely to be older than assortment on education itself. The genetic partner correlations could be predicted from the genetic sibling

correlations and vice versa. The results thus indicate that assortment on genetic factors related to educational attainment has occurred long enough for it to be in or close to equilibrium. Results from the polygenic scores and the rGenSi model both supported this. Genetic variance close to equilibrium can be observed after only five generations (see Supplementary Figure 6). Hence, one could still observe the present results if assortment on variables related to educational attainment started as late as in the middle of the 19th century. Our finding of genetic equilibrium in educational attainment aligns with stable genetic spouse similarity among individuals born in the first half of the 20th century [31] but contrasts Kong et al.'s [33] finding that genetic educational assortment is a recent phenomenon in Iceland. This contrast may be related to differences in methods, such as our use of siblings and in-laws rather than only partners, and calls for further replication. Alternative versions of our model, presented in the Supplementary Table 4, indicated deviations from equilibrium, but those versions of the model made unrealistic assumptions such as direct assortment and had poor fit. If our results are correct, we do not expect changes in the genetic variance of educational attainment [18] and related health outcomes [15] unless the level of assortment changes. For height and depression, the results also did not indicate deviations from intergenerational equilibrium.

The discussion of direct vs indirect assortment on page 9 line 21 to page 10 line 15 is now also relevant for this finding (copied under our respond to this reviewer's comment #1 above).

19) P. 21 – table 1 – the “Phenotype x Polygenic score” nomenclature is confusing. It’s not clear what this means. I was thinking interaction when I saw it, but it’s $r(Y, PGS)$.

RESPONSE: Thank you for pointing this out. We reported correlations between the phenotype and the polygenic score. The table has been updated accordingly.

20) P. 23 – fig 2 – seems too simple to put in its own figure. It’s already in fig 3 anyway.

RESPONSE: The figure has been removed from the manuscript.

21) P. 24 – fig 3 – the residual terms shouldn’t all use the same letter (“R”)

RESPONSE: This has been fixed in the revised version of the model/manuscript.

22) P. 25 – fig 4 – the lines between the observed r and the $r_{genetic}$ signals is misleading (as if it were increasing), but the two are measuring different things.

RESPONSE: We no longer use this figure in the revised manuscript. The new Figure 4 provides similar information.

23) P. 25 – fig 4 – needs to be explicit that this is from the extended rGenSi model (if the authors continue to use both models in the MS).

RESPONSE: This comment is no longer relevant as we only use a model of the extended family.

Reviewers' Comments:

Reviewer #1:

Remarks to the Author:

I thank the authors for addressing most of my comments. The manuscript reads much better although I think there are still a few things to clarify. I list those below

Major comments

Model specification. The model used by the authors uses a latent genetic factor (let's denote it g) that can be expressed as $g = PGS + E_g$, where E_g contains estimation errors and parts of the genetic signal not captured by SNPs included in the score. I'm not familiar with the implementation of the rGenSi model but it seems like the model implicitly assumes that the PGS is independent of E_g . However, we would expect the PGS and E_g to be correlated under assortative mating (AM). Can the authors include a parameter accounting for such correlation in their model and check if their conclusions change? I suspect that it may affect conclusions about inter-generational equilibrium.

Inter-generational equilibrium. The authors now compare their observed estimates with predictions under equilibrium conditions. This is a nice addition to the paper, which actually contradicts the conclusions made in the previously submitted version. I am not quite sure how valid these predictions are when the assortment is not directly on the phenotype of interest as reported here for educational attainment (EA) and major depression disorder (MDD). If that theory does not apply then the conclusions may not be correct either. I urge the authors to address this point.

Statistical significance. I found the manuscript unclear about what is considered to be a significant finding or not. The authors use 95% confidence intervals (CI), which implies that their implicit significant threshold is 0.05. Are the authors accounting for multiple testing anyhow? As the study explores 3 traits, it may be more appropriate to use 0.05/3 as significant threshold. Either way, clearly stating the significance threshold is critical to determine the robustness of the certain claims. That was the essence of my previous comment about MDD. The reported correlation from the rGenSi model is 0.08 with a 95%CI equal to [0.03 - 0.11]. Assuming that this CI is based upon the asymptotic normality distribution of maximum likelihood estimators then that would imply a standard error of $(0.11-0.03)/(2 \times 1.96) = 0.02$ and a p-value (under the same assumptions) of $9e-5$, which is quite significant.

Minor

These a few suggestions to improve the language hence clarity of the manuscript.

General comments

Sometimes the authors use "related" to mean "correlated". I'd suggest using "correlated" given that related can be confusing here as you analyse relatives.

Sorry I did not flag this before but I think it will be more appropriate and specific to use "Major Depression Disorder (MDD)" instead of "depression". I know that they authors clearly mention that in the methods but I believe the main text would read better if MDD was used throughout.

Specific comments

Abstract. R_g for depression should be 0.08 (best model) and not 0.03, right?

Abstract. It could help to use "N=" to guide the reader. For example: " pairs of partners (N=26,681), ...".

L53-54. "Increased genetic variation is followed by a larger variation between individuals in phenotypic expression." The use of "followed" in this context is incorrect because the increase of genetic and phenotypic variance happens simultaneously. I suggest, "Increased genetic variation translates into larger inter-individual phenotypic differences."

L93 – 94. Rawlik et al. (2019) [<https://www.nature.com/articles/s41437-019-0185-3>] would be a relevant reference to add there.

L124. "lets us separate" => "allows us to separate" sounds better.

L136 – 149: should say that no significant correlation was observed for MDD. Also the first part the paragraph that describes the table is not needed. It should be moved to Table legend.

L152 (and possibly somewhere else): "this parameter is an estimate". This does not sound right from a statistical point of view. I suggest "this parameter measures ..."

L155. "different versions". Could you please clarify what these versions are in the main text? These versions corresponds to constrains on the model parameters. Or alternatively there are different "models". Just need to clarify that for the reader.

L259. The authors should acknowledge that many more factors could explain why their findings about educational attainment differ from Kong et al. For example, statistical power (no evidence for something does not mean evidence of the absence) or simply difference between populations. AM is a behavioral/cultural trait and behaviors vary between countries and populations.

Reviewer #2:

Remarks to the Author:

I was impressed by the last submission although I felt it needed work for clarity. I'm even more impressed by this paper, which is not just clearer but the science is better. The authors have done a tremendous amount of work in the resubmission, basically creating a new and more realistic model, assuring themselves it works in simulated data, and testing it in the MoBa dataset. Kudos to the authors. These findings are interesting and important. I think the results should be published in Nature Communications, but I still believe the authors should clear some things up, which I list below.

GENERAL POINTS:

1) I think the authors' model 2 (including a and c) is a nice addition because it makes the model more realistic for many traits, including EA and depression. However, the authors need to discuss that the model is leaving out an important process (for EA at least)—genetic nurture and therefore vertical transmission. Kong et al and the EA3 GWAS have shown that G-E covariance is an important component of EA GWAS sumstats (e.g., ~50%) and therefore of PGSs. However, this isn't modeled by the authors. c^2 assumes that environments are shared in siblings but not passed down intergenerationally and therefore that there is no passive covariance between the genetic scores (and the PGSs) and the shared sibling environment. I personally think the authors have gone far enough – no model is perfect! But I do think it is important for them to name this as a limitation and to discuss what the effect of such G-E covariance might have on their results given that their models omit it.

2) In the previous review, I had said that "The PGS scores already have the influences of genetic nurture and AM embedded within them. What influence does this have on the authors' results?" I'm not sure I am convinced by the authors' response, which is about how these effects would influence the PGSs of all relatives the same. True, but the estimates from this model do not solely rely on the

ratios of covariances between relatives. The absolute covariances go into many of these estimates. It should be simple enough for the reviewers to simulate data where PGSs are already inflated by AM, and then see what influence this has on estimates.

This is a slightly different issue than above. One could model vertical transmission and genetic nurture (the above issue), but still use PGSs that are inflated by indirect genetic effects, and that still might have some influence on the model.

SPECIFIC POINTS, MAIN TEXT:

p. 11 – The explanation for why the conclusion about AM being at equilibrium for EA is different for this study than the Kong study seems insufficient. The thought that it's related to differences in methods seems like hand-waving. The Kong method and the present one really rely on roughly the same information: the correlation of haplotypic PGSs within-person (Kong) vs. the correlation of PGSs across siblings (this one), which are the same bits of information when one considers that within-parent haplotypic PGS correlations would create genetic correlations $> .50$ for siblings of those parents. Put another way: consider generations G0 (gen 0, the offspring gen); G1 (parental gen), G2, ...G_{inf}. Kong included participants from G0 and G1. This study only includes participants from G1 (although that difference isn't relevant here as shown below):

rg_sib (this study) is affected by AM from G2, G3, ...G_{inf}

rg_spouse (this study) is affected by AM from G1

Similarly,

Kong rg_haplotypes_within_parent is affected by AM from G2, G3, ...G_{inf}

Kong rg_spouse is affected by AM from G1

The rg_sib and rg_haplotypes are using the same information regarding gametic phase disequilibrium arising from G2, G3... G_{inf}, so I don't see how a difference in methods would lead to differing conclusions if they've both been done correctly. It seems more likely that this might be due to a difference in culture. It's also possible that, if the Kong parents are on average older than the MoBa parents, which could explain the discrepancy. E.g., say that Kong parents were G2 whereas MoBa parents are G1, and that AM started in G2 – this could lead to the pattern of results observed (in addition to the next point about power). In summary, can the authors discuss the possible reasons for the discrepancy with a bit more thought about possible reasons?

p. 11/p. 18 – a minor point but related to the above, most of the effect of AM on rg_sib occurs in one generation of AM, and about 60% of it is determined by the previous 2 generations of AM. There's going to be little power in differentiating whether rg_sib is due to AM from previous 1 or 2 generations vs. many required to reach equilibrium. This should probably be stated somewhere so that readers don't wrongly assume that this shows that AM has been going on indefinitely for these traits.

p. 12 & p. 16 – the Kong study adjusted for 100 PCs. This one only 20. That's also a possible cause for the discrepancy that the authors should investigate and eliminate if possible.

p. 13 – four key findings paragraph – for the first one, it needs to be clear that this applies to EA and Dep only; for height, the phenotypic and genetic correlations agree.

P. 17 – Authors state that expected correlations can be found using standard path tracing rules, but this may be confusing to anyone knowledgeable about SEM because copaths are not standard. Perhaps authors should be a bit clearer about the copath (m) rules, given that the path tracing they describe breaks normal path tracing rules (e.g., that the paths cannot change directions more than

once).

p. 18 – Authors state "The residual correlation between siblings' polygenic scores can be estimated freely if effects of shared environment (c) are not included in the model but is 0.50 in realistic scenarios." I tripped over this at first but now I think I get it. Nevertheless, it seems technical and could lead to confusion – perhaps move it to the Supp. where you can explain it a bit more?

The Fig 3 caption should explain the "fixed" vs. "free" distinction a bit

Why does the final figure of the MS show $a=1$, as though it is fixed. This is estimated, no?

SPECIFIC POINTS, SUPPLEMENTS:

In the Supp Fig 3, Eg is the error component of the PGS, unrelated to the phenotype and thus unaffected by AM, and this is why it is correlated at .50 bw siblings (rather than higher), right? Perhaps this point could be made a bit clearer somewhere.

In the Supp (p. 6) authors state that "The correlation between one's polygenic score and one's own phenotype is $s * h$, whereas the correlation between one's polygenic score and one's partner's phenotype is $s * h * m$. Hence, m can be estimated by dividing the cross-partner phenotype-polygenic correlation with the within individual phenotype-polygenic correlation." Didn't the authors leave out an alpha in these? I.e., the first should equal $s*h*a$ and the latter $s*h*m*a$?

In the simulations described in the Supp, the sample sizes were 1K and 50K. These were individuals, correct? And how were they distributed across relative types? Were all extended families complete (i.e., 2 sibs & 2 spouses), such that for $n=1K$, there were 250 complete extended families?

In the Supp, the way that authors simulated AM such that a given spousal correlation was maintained needs to be explained, as this can be challenging in a forward-time simulation.

In the Supp p. 7, the authors describe how they drew a and c^2 , but not the other parameters. Am I right that h , m , and s can all be freely drawn in a simulation, whereas r is determined by the other parameters?

- Matthew Keller

REVIEWER COMMENTS

Reviewer #1 (Remarks to the Author):

I thank the authors for addressing most of my comments. The manuscript reads much better although I think there are still a few things to clarify. I list those below

Major comments

[Comment 1:] Model specification. The model used by the authors uses a latent genetic factor (let's denote it g) that can be expressed as $g = \text{PGS} + E_g$, where E_g contains estimation errors and parts of the genetic signal not captured by SNPs included in the score. I'm not familiar with the implementation of the rGenSi model but it seems like the model implicitly assumes that the PGS is independent of E_g . However, we would expect the PGS and E_g to be correlated under assortative mating (AM). Can the authors include a parameter accounting for such correlation in their model and check if their conclusions change? I suspect that it may affect conclusions about inter-generational equilibrium.

RESPONSE: In the model, g and E_g are both latent components of variance in a polygenic score, so while $\text{PGS} = g + E_g$, the reviewer's expression of the model (" $g = \text{PGS} + E_g$ ") is not correct. Below, we show two hypothetical models.

The model to the left represents a possible true model, whereas the model to the right represents what we have fitted to the data for each person. To the left g is split in two components, one part measured by the PGS (A) and one part not measured by the PGS (B). By definition, g and E_g are orthogonal. ("The genetic residual is defined as not being associated with the phenotype", line 464-461.) This means that the model does not assume that " E_g contains [...] parts of the genetic signal not captured by SNPs included in the score". The covariance between PGS and E_g can simply be derived by following path tracing rules, which means that it equals $e = \text{sqrt}(1 - s^2)$. However, potential correlation between A and B is a more important concern. Under AM, A and B can become correlated r . A positive correlation r would not influence our results. E_g is defined as not being related to the phenotype, and is hence unrelated to both A and B , and hence also unrelated to G . There are no valid paths connecting E_g to A , B , or to G , hence the correlations between these are 0. This consideration is already a part the simulation, as the data were generated similarly to the figure to the left and we show in the supplement page 3-11 that the parameters of interest for our study

could be estimated with the model to the right. In the discussion, page 12 line 296 – page 13 line 320, we write:

Ideally, the latent genetic factor should be modelled as the sum of two components – trait-increasing alleles included in the polygenic score (A) plus trait-increasing alleles not included in the polygenic (B) score [35]. The polygenic score itself should then be modelled as the sum of trait-increasing alleles (A) and measurement error. This would have enabled us to distinguish measurement error from partialness of the polygenic scores. However, there were not enough degrees of freedom in the present data, and estimating the resulting correlations was sufficient to reach our objectives.

The section “Testing the assumption that the residual genetic correlation between siblings is 0.50” may also be relevant on page 8-9 of the supplement.

[Comment 2:] Inter-generational equilibrium. The authors now compare their observed estimates with predictions under equilibrium conditions. This is a nice addition to the paper, which actually contradicts the conclusions made in the previously submitted version. I am not quite sure how valid these predictions are when the assortment is not directly on the phenotype of interest as reported here for educational attainment (EA) and major depression disorder (MDD). If that theory does not apply then the conclusions may not be correct either. I urge the authors to address this point.

RESPONSE: We thank the reviewer for this comment and have now included a more thorough discussion and simulation of this in the supplement. In brief, our approach to testing intergenerational equilibrium is based on comparing genetic correlations between siblings and partners. Since these are estimated as correlations between latent variables, they are not affected by imperfect assessment of polygenic scores or of phenotypes (the a and s are not included in valid paths connecting relatives’ latent genetic variables in Figure 2). Indirect selection is indicated by an estimated value of $a < 1$. Similarly, imperfection of polygenic scores is indicated by $s < 1$. We have shown in the supplement that $s < 1$ underestimates to the same degree partner correlations (toward 0.00) and sibling correlation (towards 0.50). We have now extended this discussion and included a simulation showing that the results are robust to low values of a , whereas low values of s lead to more uncertainty, although not bias in any particular direction. Please see page 9-10 in the supplement, section “Testing intergenerational equilibrium with imperfect indicators of genotype and phenotype”

[Comment 3:] Statistical significance. I found the manuscript unclear about what is considered to be a significant finding or not. The authors use 95% confidence intervals (CI), which implies that their implicit significant threshold is 0.05. Are the authors accounting for multiple testing anyhow? As the study explores 3 traits, it may be more appropriate to use 0.05/3 as significant threshold. Either way, clearly stating the significance threshold is critical to determine the robustness of the certain claims. That was the essence of my previous comment about MDD. The reported correlation from the rGenSi model is 0.08 with a 95%CI equal to [0.03 – 0.11]. Assuming that this CI is based upon the asymptotic normality distribution of maximum likelihood estimators then that would imply a standard error of $(0.11-0.03)/(2 \times 1.96) = 0.02$ and a p-value (under the same assumptions) of $9e-5$, which is quite significant.

RESPONSE: We agree that the criteria statistical significance should be clearly defined and apologize for not being clear about this. We perform null hypothesis significance tests with a 5% alpha level, with the null value set to either 0.00 or 0.50 depending on the parameter in question. That is, when we are testing whether sibling correlations are inflated, we use 0.50 as the null, and in all other situations we test against a null value of 0.00. We have not adjusted for multiple testing, as the

hypotheses are considered independent, i.e., we do not take a given result in any domain to make general conclusions across domains, and because the false-positive rate is known to the reader. In response to this comment, we have updated the methods section with information on hypothesis testing and statistical significance on page 17 line 414-416.

We provide 95% confidence intervals for all estimates and perform null hypothesis significance tests with a 5% alpha level, with the null value set to either 0.00 or 0.50 depending on the parameter in question.

We have also updated the results section concerning MDD at page 6 line 139-147 and the test of intergenerational equilibrium at page 6 line 139-146.

We consider associations to be significantly different from 0.00 or 0.50 when the 95% confidence intervals do not include these numbers. For all three phenotypes, there were positive correlations between partners, siblings, in-laws, and co-in-laws. All three polygenic scores were robustly associated with the individuals' own phenotype, the phenotype of their sibling, and the phenotype of their partner. For educational attainment, the polygenic score was also associated with in-laws' ($r=0.11$, 95%CI 0.09, 0.13) and co-in-laws' ($r=0.08$, 95%CI 0.05, 0.11) educational attainment. The polygenic scores were correlated between partners for educational attainment ($r=0.11$, 95%CI 0.10, 0.12) and height ($r=0.05$, 95%CI 0.03, 0.06) but not depression ($r=0.00$, 95%CI -0.01, 0.01).

Minor

These a few suggestions to improve the language hence clarity of the manuscript.

General comments

[Comment 4:] Sometimes the authors use “related” to mean “correlated”. I’d suggest using “correlated” given that related can be confusing here as you analyse relatives.

RESPONSE: We have replaced “related to” with “correlated with” or “associated with” several places in the manuscript to avoid this confusion. The changes are highlighted in yellow in the revised manuscript.

[Comment 5:] Sorry I did not flag this before but I think it will be more appropriate and specific to use “Major Depression Disorder (MDD)” instead of “depression”. I know that they authors clearly mention that in the methods but I believe the main text would read better if MDD was used throughout.

RESPONSE: We thank the reviewer for pointing out the distinction between major depressive disorder (MDD) and depression more generally, which we have now attempted to clarify in the manuscript. We fear that the use of the term major depressive disorder (MDD) may signal the use of diagnostic data, which we did not have available. Our phenotypic measure was intended to measure symptoms of MDD. At the same time the depression GWAS [1] includes both MDD and broader depression measures. This distinction is probably not of high importance for genetic studies, as “Howard, et al. showed that there is a strong genetic correlation ($r_G = 0.86$, s.e. = 0.05) between broader self-declared definitions of depression and clinically diagnosed major depressive disorder (MDD) within a hospital setting” (cited from [1]).

Since we do not have a diagnostic MDD measure and the GWAS is not exclusively based on MDD, we continue to use the term “depression” for short. Previously, this was not defined until the Methods

section, but it is now specified in the introduction, so the contents of the measure are clear for the reader from the beginning. Please see changes the following changes:

Introduction on page 4 line 121:

We investigate assortment on educational attainment, height, and depression (symptoms of major depressive disorder), [...]

Methods page 16 line 389-399:

The genetic association between major depressive disorder and broader definitions of depression has been reported to be very high ($r=0.86$) [54].

Specific comments

[Comment 6:] Abstract. R_g for depression should be 0.08 (best model) and not 0.03, right?

RESPONSE: We thank the reviewer for pointing this out. We have updated the abstract.

[Comment 7:] Abstract. It could help to use “N=” to guide the reader. For example: “ pairs of partners (N=26,681), ...”.

RESPONSE: We thank the reviewer for this suggestion and have changed the abstract accordingly.

[Comment 8:] L53-54. “Increased genetic variation is followed by a larger variation between individuals in phenotypic expression.” The use of “followed” in this context is incorrect because the increase of genetic and phenotypic variance happens simultaneously. I suggest, “Increased genetic variation translates into larger inter-individual phenotypic differences.”

RESPONSE: We thank the reviewer for this suggestion and have updated the manuscript accordingly.

[Comment 9:] L93 – 94. Rawlik et al. (2019) [<https://www.nature.com/articles/s41437-019-0185-3>] would be a relevant reference to add there.

RESPONSE: We thank the reviewer for the reference to this relevant paper. It is now included and cited as reference 25 on page 4 line 93.

[Comment 10:] L124. “lets us separate” => “allows us to separate” sounds better.

RESPONSE: This has been updated according to the reviewer’s suggestion.

[Comment 11:] L136 – 149: should say that no significant correlation was observed for MDD. Also the first part the paragraph that describes the table is not needed. It should be moved to Table legend.

RESPONSE: We have updated this paragraph. The sentence describing Table 1 has been moved to the legend of that table. Concerning depression/MDD, page 6 line 141-443 now states:

All three polygenic scores were robustly associated with the individuals’ own phenotype, the phenotype of their sibling, and the phenotype of their partner.

Page 6 line 145-147 states:

The polygenic scores were correlated between partners for educational attainment ($r=0.11$, 95%CI 0.10, 0.12) and height ($r=0.05$, 95%CI 0.03, 0.06) but not depression ($r=0.00$, 95%CI -0.01, 0.01).

[Comment 12:] L152 (and possibly somewhere else): “this parameter is an estimate”. This does not sounds right from a statistical point of view. I suggest “this parameter measures ...”

RESPONSE: We have changed this to according to the reviewer’s suggestion on page 7 line 152 and in the following paragraph on page 7 line 166.

[Comment 13:] L155. “different versions”. Could you please clarify what these versions are in the main text? These versions corresponds to constrains on the model parameters. Or alternatively there are different “models”. Just need to clarify that for the reader.

RESPONSE: We have updated the text on page 7 line 155-156.

The results of fitting to the data *rGenSi* models with a either freely estimated or fixed to 1.00 are presented in Supplementary Table 5. The estimated parameters of these different versions of the model are presented in Supplementary Table 6.

[Comment 14:] L259. The authors should acknowledge that many more factors could explain why their findings about educational attainment differ from Kong et al. For example, statistical power (no evidence for something does not mean evidence of the absence) or simply difference between populations. AM is a behavioral/cultural trait and behaviors vary between countries and populations.

RESPONSE: We agree that this was too brief. We now discuss differences between our results and Kong et al.’s results in more detail on page 11 line 259 – page 12 line 277:

The genetic variance increases fastest in the first generations with assortment. Genetic variance close to equilibrium can be observed after only five generations, whereas 60% of the increase in variance is seen after two generations (see Supplementary Figure 6). Hence, one could observe approximate equilibrium if assortment on variables associated with educational attainment started as late as the end of the 19th century. The larger part of the distance to equilibrium can also be covered for even newer phenomena. Our finding of genetic equilibrium in educational attainment could therefore be expected from century old descriptions of partner similarity in related traits [8, 10] and aligns with stable genetic spouse similarity among individuals born in the first half of the 20th century [5]. It does, however, contrast with Kong et al.’s [35] finding that genetic educational assortment is a recent phenomenon in Iceland. The educational system in Iceland was developed later than in Norway, from which our sample comes. Unlike Norway, most areas of Iceland had few or no schools in the 19th century [42] and the first university was founded 100 years later than the first in Norway (1911 vs. 1811). Hence, it is possible that education became relevant as an assortment factor later in Iceland than in Norway. A weakness with this explanation is that Icelanders could have chosen partners based on factors that are related to education today even with low access to formal education. Rurality and unidentified cultural differences are therefore alternative explanations. In addition, we cannot exclude that the contrast in results is related to differences in methods, such as our use of siblings and in-laws rather than only partners. This finding therefore calls for further replication.

Reviewer #2 (Remarks to the Author):

[Comment 15:] I was impressed by the last submission although I felt it needed work for clarity. I'm

even more impressed by this paper, which is not just clearer but the science is better. The authors have done a tremendous amount of work in the resubmission, basically creating a new and more realistic model, assuring themselves it works in simulated data, and testing it in the MoBa dataset. Kudos to the authors. These findings are interesting and important. I think the results should be published in Nature Communications, but I still believe the authors should clear some things up, which I list below.

RESPONSE: We thank the reviewer for a positive evaluation and are grateful for insightful comments.

GENERAL POINTS:

[Comment 16:] 1) I think the authors' model 2 (including a and c) is a nice addition because it makes the model more realistic for many traits, including EA and depression. However, the authors need to discuss that the model is leaving out an important process (for EA at least)—genetic nurture and therefore vertical transmission. Kong et al and the EA3 GWAS have shown that G-E covariance is an important component of EA GWAS sumstats (e.g., ~50%) and therefore of PGSs. However, this isn't modeled by the authors. c^2 assumes that environments are shared in siblings but not passed down intergenerationally and therefore that there is no passive covariance between the genetic scores (and the PGSs) and the shared sibling environment. I personally think the authors have gone far enough – no model is perfect! But I do think it is important for them to name this as a limitation and to discuss what the effect of such G-E covariance might have on their results given that their models omit it.

RESPONSE: Our model assumed independence of the latent genetic factor (G), shared environment (C), and non-shared environment (E). We now note this as a limitation of the model, please see page 13 line 314-319.

Second, the *rGensi* model makes certain simplifying assumptions. Extensive simulations indicated that the model provided the expected results and the empirical results that could be compared with previous studies were mainly in line with these. However, we did not include covariance between an individual's latent genetic and environmental factors, which would be expected under vertical transmission. Assuming independence is a limitation of the current model and it remains to be determined how this influences the results in various scenarios.

Vertical transmission is documented for educational attainment, but it is unclear to what degree this is relevant under indirect assortment. Out of curiosity, we simulated data with vertical transmission, to determine how this might influence the results when it is assumed to be zero. For instance, we generated a dataset (N=50,000) with $h^2=0.40$, $m=0.50$, $c^2=0.10$, $s=1.00$, and $a=1.00$ (starting values first generation, all values freely estimated at the end). In this simulation, c is entirely due to vertical transmission of the parental phenotype. After 10 generations, we estimated the *rGenSi* model. This had little impact on the estimated genetic resemblance between relatives, which depend on h , m , and r , and not on c , but increased estimated correlations somewhat. (Partners: Observed $r_g=0.31$ estimated $r_g=0.35$; siblings: observed $r_g=0.66$, estimated $r_g=0.69$; inlaws: observed $r_g=0.22$, estimated $r_g=0.25$; coinlaws: observed $r_g=0.08$, estimated $r_g=0.09$). This may be different in other scenarios. We have added an option VT=[FALSE|TRUE] to the simulation script (default VT=FALSE).

[Comment 17:] 2) In the previous review, I had said that "The PGS scores already have the influences of genetic nurture and AM embedded within them. What influence does this have on the authors' results?" I'm not sure I am convinced by the authors' response, which is about how these effects would influence the PGSs of all relatives the same. True, but the estimates from this model do not solely rely on the ratios of covariances between relatives. The absolute covariances

go into many of these estimates. It should be simple enough for the reviewers to simulate data where PGSs are already inflated by AM, and then see what influence this has on estimates.

This is a slightly different issue than above. One could model vertical transmission and genetic nurture (the above issue), but still use PGSs that are inflated by indirect genetic effects, and that still might have some influence on the model.

RESPONSE: As we understand this, the reviewer is concerned that AM (in previous generations) have inflated the covariance between the phenotype and the polygenic score, i.e., not only does AM lead to larger genetic variation (higher heritability), but that it might also bias results from GWAS such that the heritability is estimated to be even higher than what it really is, because regression weights are inflated (or in the case of cross-trait AM inflated above 0). If the heritability estimates are overestimated (in line with Border et al., 2021 [2]), this will be expected to lead to higher genetic correlations between relatives in our model, because these depend (to a large degree) on the h parameter.

We simulated according to the description in the supplementary information with $h^2=0.30$, $m=0.50$, $c^2=0.00$, $s^2=1.00$, and $a=1.00$ (starting values, h^2 increases after generations with assortment). We then increased the correlation between the phenotypes and the polygenic scores (but not the latent genetic variables). We then inflated the phenotypic variance explained by polygenic scores with 50%. We estimated our model after 10 generations. Expectedly, this led to a higher heritability estimate (62% instead of 40%, ~50% higher). Other parameters were intact. As the correlation between relatives depend on h , they were inflated too (partners: Observed $r_g=0.22$ estimated $r_g=0.31$; siblings: observed $r_g=0.59$, estimated $r_g=0.56$; inlaws: observed $r_g=0.09$, estimated $r_g=0.17$; coinlaws: observed $r_g=0.02$, estimated $r_g=0.05$). These results are entirely as expected for an overestimated heritability (one could get the same results by following path tracing rules in the rGenSi model and simply using an inflated value for h). We believe that the inflation of 50% is too high for a realistic scenario, it is used here to show a worst-case scenario. We are, however, not sure which rate of inflation that is realistic. The field is currently in the process of finding out this may influence GWAS results. Several recent papers/preprints deal with related topics, trying to understand how assortative mating influences heritability estimates and to what degree within-family (sibling) analyses can counteract these problems [2-4]. We believe this topic needs further exploration in the future. We discuss this as a limitation in the study. As we base our polygenic scores on the most recent GWAS, the amount of bias in our polygenic scores should be comparable to other studies in the field. Please see the limitations on page 13 line 320-324:

Assortative mating and parental indirect genetic effects may have influenced these studies and thereby also inflated our estimates of heritability. If we have overestimated heritability, genetic correlations between relatives will most likely also be inflated. Recent studies are exploring how these phenomena influence genomic studies and how to counteract this [44-46].

SPECIFIC POINTS, MAIN TEXT:

[Comment 18:] p. 11 – The explanation for why the conclusion about AM being at equilibrium for EA is different for this study than the Kong study seems insufficient. The thought that it's related to differences in methods seems like hand-waving. The Kong method and the present one really rely on roughly the same information: the correlation of haplotypic PGSs within-person (Kong) vs. the correlation of PGSs across siblings (this one), which are the same bits of information when one considers that within-parent haplotypic PGS correlations would create genetic correlations $> .50$ for siblings of those parents. Put another way: consider generations G_0 (gen 0, the offspring gen); G_1 (parental gen), G_2 , ... G_{inf} . Kong included participants from G_0 and G_1 . This study only includes

participants from G1 (although that difference isn't relevant here as shown below):

rg_sib (this study) is affected by AM from G2, G3, ...Ginf

rg_spouse (this study) is affected by AM from G1

Similarly,

Kong rg_haplotypes_within_parent is affected by AM from G2, G3, ...Ginf

Kong rg_spouse is affected by AM from G1

The rg_sib and rg_haplotypes are using the same information regarding gametic phase disequilibrium arising from G2, G3... Ginf, so I don't see how a difference in methods would lead to differing conclusions if they've both been done correctly. It seems more likely that this might be due to a difference in culture. It's also possible that, if the Kong parents are on average older than the MoBa parents, which could explain the discrepancy. E.g., say that Kong parents were G2 whereas MoBa parents are G1, and that AM started in G2 – this could lead to the pattern of results observed (in addition to the next point about power). In summary, can the authors discuss the possible reasons for the discrepancy with a bit more thought about possible reasons?

RESPONSE: We acknowledge that this was too brief and have expanded the discussion on this topic. Kong et al. notes that “assortative mating occurring before the parents’ generation could lead to additional confounding. However, this effect appears to be negligible in our study” (p. 3). Studies describe assortative mating based on education or related factors a century ago. For instance, Conley et al. described that stable educational assortment over the 20th century with regard to genetic factors, and Jones (1929) describes spouse correlations in intelligence. One could therefore expect that the genetic consequences should have partially or fully materialised by now. These studies are, however, not from Iceland, from which Kong et al. draw their sample. We have researched the educational history of Iceland and Norway, where our sample comes from, to identify any possible differences.

As Iceland is more rural and remote than Norway, variation in education may have become relevant as a sorting mechanism later in Iceland than in Norway. This may potentially explain differences in the populations regarding equilibrium. As an example, the University of Oslo, Norway, was founded in 1811, whereas the University of Iceland was founded a full century later, in 1911. (A full century allows going from no mating to almost equilibrium.) Large parts of Norway are also close to Denmark, where a university was established in 1479. Of course, the universities served only a small number of students in their initial years, but this may reflect the, in general, later build-up of the educational system in Iceland. We have not been able to find a thorough description of the history of education in Iceland, however, Tveit (1991) [5] writes this: *“In Iceland writing proficiency was made compulsory in 1888. Since there were at that time no schools in rural districts, and only a few in the towns, instruction in writing had to be conducted in the homes, as had been the case with reading.”* On the other hand, training in reading were made mandatory in Norway in 1736, and in 1889 it was made mandatory to attend primary school for 7 years. Hence, in 19th century Norway there were schools available and at least one university. In sum, education as a factor for assortment may have started later in Iceland than in Norway. It is plausible the variation between individuals in education became a relevant factor for assortment earlier in Norway than in Iceland, and hence, that Norway has move further towards the intergenerational equilibrium. A weakness with this explanation is that Icelanders could have chosen partners based on factors that are related to education today (intelligence, conscientiousness) even without having formal education. Therefore, we cannot conclude that this is the reason of the different findings.

MoBa parents were born, on average, in 1972, whereas Kong et al.'s participants were born between 1940 and 1983 – if averaged across this time span, the average birth year is 1962. We do not, however, expect these 10 years to make a large difference.

Finally, there could be differences in the methods that we have not yet identified.

We have updated the section on differences between our study and Kong et al.'s study.

Discussion on page 11 line 259 – page 12 line 277:

The genetic variance increases fastest in the first generations with assortment. Genetic variance close to equilibrium can be observed after only five generations, whereas 60% of the increase in variance is seen after two generations (see Supplementary Figure 6). Hence, one could observe approximate equilibrium if assortment on variables associated with educational attainment started as late as the end of the 19th century. The larger part of the distance to equilibrium can also be covered for even newer phenomena. Our finding of genetic equilibrium in educational attainment could therefore be expected from century old descriptions of partner similarity in related traits [8, 10] and aligns with stable genetic spouse similarity among individuals born in the first half of the 20th century [5]. It does, however, contrast with Kong et al.'s [35] finding that genetic educational assortment is a recent phenomenon in Iceland. The educational system in Iceland was developed later than in Norway, from which our sample comes. Unlike Norway, most areas of Iceland had few or no schools in the 19th century [42] and the first university was founded 100 years later than the first in Norway (1911 vs. 1811). Hence, it is possible that education became relevant as an assortment factor later in Iceland than in Norway. A weakness with this explanation is that Icelanders could have chosen partners based on factors that are related to education today even with low access to formal education. Rurality and unidentified cultural differences are therefore alternative explanations. In addition, we cannot exclude that the contrast in results is related to differences in methods, such as our use of siblings and in-laws rather than only partners. This finding therefore calls for further replication.

[Comment 19:] p. 11/p. 18 – a minor point but related to the above, most of the effect of AM on rg_sib occurs in one generation of AM, and about 60% of it is determined by the previous 2 generations of AM. There's going to be little power in differentiating whether rg_sib is due to AM from previous 1 or 2 generations vs. many required to reach equilibrium. This should probably be stated somewhere so that readers don't wrongly assume that this shows that AM has been going on indefinitely for these traits.

RESPONSE: This is correct, Supplementary Fig. 6 shows that approximately 60% of the increase in genetic variance comes in the two first generations of AM. We have now updated the discussion on page 11 line 259-262:

The genetic variance increases fastest in the first generations with assortment. Genetic variance close to equilibrium can be observed after only five generations, whereas 60% of the increase in variance is seen after two generations (see Supplementary Figure 6).

[Comment 20:] p. 12 & p. 16 – the Kong study adjusted for 100 PCs. This one only 20. That's also a possible cause for the discrepancy that the authors should investigate and eliminate if possible.

RESPONSE: We have added results adjusting for 50 principal components to Supplementary Table 9. The results were very similar to those adjusting for 20 principal components. As the 50 first principal components had little influence on the results, it is implausible that additional principal components

should fundamentally change the results. We were unable to construct 100 principal components due to computational limitations.

[Comment 21:] p. 13 – four key findings paragraph – for the first one, it needs to be clear that this applies to EA and Dep only; for height, the phenotypic and genetic correlations agree.

RESPONSE: We did not (intend to) compare phenotypic and genetic correlations in this first finding, but rather to compare correlations between polygenic scores with correlations between underlying genetic factors, estimated with structural equation models. Hence, we have rewritten the referred sentence to make this clearer. Also, we have included “may” in the sentence as we do not think that polygenic scores must always correlate lower, but that they may do so. Please see page 14 line 334-337:

First, correlations between relatives’ polygenic scores may reflect only a part of the genetic similarity arising from assortative mating. We have presented a structural equation modelling approach to estimate genetic correlations between individuals when phenotypic data is available.

[Comment 22:] P. 17 – Authors state that expected correlations can be found using standard path tracing rules, but this may be confusing to anyone knowledgeable about SEM because copaths are not standard. Perhaps authors should be a bit clearer about the copath (m) rules, given that the path tracing they describe breaks normal path tracing rules (e.g., that the paths cannot change directions more than once).

RESPONSE: We now briefly describe copaths in the manuscript and in the supplement.

Please see the manuscript page 18 line 439-442:

Correlations between different individuals’ genetic signals can be estimated by following path tracing rules allowing for co-paths. Co-paths connect valid chains of paths [21, 33]. They contribute to the co-variance between variables, but not their variance. They are useful for analysing assortment processes, which do not change the values of variables.

Please see the supplement page 3, second paragraph:

The model assumes that partners assort directly on the phenotype; hence partners are linked by the co-path m . All partner similarity goes through this co-path. Co-paths are an extension of conventional path tracing rules introduced by Cloninger [6]. A summary of rules concerning co-paths are found in Balbona et al. [7]. In short, co-paths have no arrows (direction), and connect other valid chains of paths. They contribute to covariance between variables, but not to variance. This is useful when analysing assortative mating, because the assortment of partners do not change their phenotypes, but leads to an association.

[Comment 23:] p. 18 – Authors state "The residual correlation between siblings’ polygenic scores can be estimated freely if effects of shared environment (c) are not included in the model but is 0.50 in realistic scenarios." I tripped over this at first but now I think I get it. Nevertheless, it seems technical and could lead to confusion – perhaps move it to the Supp. where you can explain it a bit more?

RESPONSE: We have removed this sentence from the manuscript. The following sentence on line 468 states that “The model is further explained in Supplementary Methods [...]”. Please see our response to comment #26, where we describe changes in the supplement.

[Comment 24:] The Fig 3 caption should explain the "fixed" vs. "free" distinction a bit

RESPONSE: We have updated the legend of Figure 3.

Figure 3. **Parameter estimates.** Estimates from the best fitting rGenSi models. Values in red are freely estimated and can take any value between 0.00 and 1.00. Values in blue are fixed ($a=1$ or $c=0$) in the best fitting models. Exact numbers and freely estimated values for all parameters are available in Supplementary Table 6. For height and depression, there are no effects of shared environment ($c=0$). For height, mating is fully based on the measured height ($a=1$), whereas for education and depression, couples assort on a correlated phenotype.

[Comment 25:] Why does the final figure of the MS show $a=1$, as though it is fixed. This is estimated, no?

RESPONSE: We thank the reviewer for pointing out this. It is correct that a is freely estimated. We have replaced $a=1$ with a and $c=0$ with c in Figure 2 and explain in the caption that these can be fixed in more restricted versions of the model.

SPECIFIC POINTS, SUPPLEMENTS:

[Comment 26:] In the Supp Fig 3, E_g is the error component of the PGS, unrelated to the phenotype and thus unaffected by AM, and this is why it is correlated at .50 bw siblings (rather than higher), right? Perhaps this point could be made a bit clearer somewhere.

RESPONSE: We have expanded on this in a separate paragraph on page 8-9 of the supplement:

Testing the assumption that the residual genetic correlation between siblings is 0.50

The residual component of the polygenic score is defined as not being associated with the phenotype. Therefore, we assumed that it could not be a basis for partner selection and that it should not correlate between partners. Similarly, we assume that this genetic residual was not associated with partner selection in previous generations and thereby assume that the residual genetic correlation between siblings is 0.50. However, this is not statistically necessary to make this assumption. It is also possible to test. If effects of shared environment (c) are fixed to a specific value (usually zero), it is possible to freely estimate the correlation between siblings' genetic residuals. In the *rGenSi* model, it is unfortunately not possible to simultaneously estimate both effects of shared environment and the correlation between siblings' genetic residuals. The genetic correlation between siblings' residuals was estimated very close to 0.50 in all tested simulations when we fixed $c=0$. (In simulations, we also generated half-siblings sharing 0.25 of genetic factors and a fictitious "worker bee" sibling type sharing 0.75 of genetic factors. In these cases, the correlation between genetic residuals matched the expected 0.25 and 0.75, respectively.) Hence, we conclude that the residual correlation between siblings' polygenic scores can be freely estimated if effects of shared environment (c) are not included in the model but that it is 0.50 in realistic scenarios.

[Comment 27:] In the Supp (p. 6) authors state that "The correlation between one's polygenic score and one's own phenotype is $s * h$, whereas the correlation between one's polygenic score and one's partner's phenotype is $s * h * m$. Hence, m can be estimated by dividing the cross-partner phenotype-polygenic correlation with the within individual phenotype-polygenic correlation." Didn't the authors leave out an alpha in these? I.e., the first should equal $s*h*a$ and the latter $s*h*m*a$?

RESPONSE: We thank the reviewer for pointing out this mistake. The provided information would be correct in the context of the model shown in Supplementary Figure 1. We have therefore moved the text to the paragraph describing that model on page 4 in the supplement and rewritten the referred section of the supplement on page 6. (As a is added both to the numerator and the denominator, the described division still yields m .)

Page S4:

The correlation between one's polygenic score and one's own phenotype is $s * h$, whereas the correlation between one's polygenic score and one's partner's phenotype is $s * h * m$. Hence, m can be estimated by dividing the cross-partner phenotype-polygenic correlation ($s * h * m$) with the within individual phenotype-polygenic correlation ($s * h$). (Robinson et al. derived the phenotypic correlation in an equivalently way by regressing a phenotype on the partner's genetic predictor [8].)

Page S6:

In this version of the model, where a is freely estimated, the correlation between one's polygenic score and one's own phenotype is $a * s * h$, whereas the correlation between one's polygenic score and one's partner's phenotype is $a * s * h * m$. It is still possible to estimate m by dividing the cross-partner phenotype-polygenic correlation with the within individual phenotype-polygenic correlation, as $(a * s * h * m) / (a * s * h) = m$. However, m now describes the similarity in the latent phenotype underlying assortment rather than the observed phenotypic similarity.

[Comment 28:] In the simulations described in the Supp, the sample sizes were 1K and 50K. These were individuals, correct? And how were they distributed across relative types? Were all extended families complete (i.e., 2 sibs & 2 spouses), such that for n=1K, there were 250 complete extended families?

RESPONSE: The reviewer is right. We have updated the supplementary materials on page 4, paragraph 2:

Each extended family consisted of four individuals, so for the sample size of 1,000 individuals, there were 250 extended families which included 250 complete pairs of siblings, 500 complete pairs of partners, 500 complete pairs of in-laws, and 250 complete pairs of co-in-laws.

[Comment 29:] In the Supp, the way that authors simulated AM such that a given spousal correlation was maintained needs to be explained, as this can be challenging in a forward-time simulation.

RESPONSE: The simulation generated individuals of two sexes. In each generation, we standardized the phenotype (mean=0; SD=1) and added a random normal variate with mean zero and variance $(1 - m) / m$, where m is the correlation among partners. This ranking was the basis of assortment. This procedure resulted in an expected correlation among partners equal to m . We have updated the supplement on page 4, second paragraph:

Within each generation, individuals within each sex were ranked according to the sum of their phenotype and a random component representing "assortment noise". The "assortment noise" was sampled from a normal distribution with mean zero and variance $(1 - m) / m$, where m is the correlation among partners. Individuals were then mated according to their rank, i.e., the highest scoring within sex 1 with the highest scoring within sex 2 and so on. The resulting partner correlations were stable over generations.

[Comment 30:] In the Supp p. 7, the authors describe how they drew a and c^2 , but not the other parameters. Am I right that h , m , and s can all be freely drawn in a simulation, whereas r is

determined by the other parameters?

RESPONSE: In the previous version of the supplement, we only described how h , m , and s were drawn in the description of the basic model (page 4 of the supplement). We now also provide a summary of this information in the description of the extended *rGenSi* model. The reviewer is right that r is determined by the other parameters. Please see the Supplementary material on page 7, second paragraph:

Otherwise, this simulation followed the same rules as the simulation of the basic *rGenSi* model, described on page 4 of this supplement. (We drew s^2 and h^2 from uniform distributions ranging from 0.20 to 1.00 and m from a uniform distribution ranging from 0.00 to 1.00. In the first generation r_s equals 0.50, but it can rise over generations with assortment, depending on the other parameters. We simulated 100 genetic variants related to a phenotype and assortment for 10 generations before we estimated the *rGenSi* model.)

- Matthew Keller

References

1. Howard, D.M., et al., *Genome-wide meta-analysis of depression identifies 102 independent variants and highlights the importance of the prefrontal brain regions*. *Nature Neuroscience*, 2019. **22**(3): p. 343-352.
2. Border, R., et al., *Assortative Mating Biases Marker-based Heritability Estimators*. bioRxiv, 2021.
3. Howe, L.J., et al., *Within-sibship GWAS improve estimates of direct genetic effects*. bioRxiv, 2021.
4. Kemper, K.E., et al., *Phenotypic covariance across the entire spectrum of relatedness for 86 billion pairs of individuals*. *Nat Commun*, 2021. **12**(1): p. 1050.
5. Tveit, K., *The Development of Popular Literacy in the Nordic Countries. A Comparative Historical Study*. *Scandinavian Journal of Educational Research*, 1991. **35**(4): p. 241-252.
6. Cloninger, C.R., *Interpretation of intrinsic and extrinsic structural relations by path analysis: theory and applications to assortative mating*. *Genetical Research*, 1980. **36**(2): p. 133-145.
7. Balbona, J.V., Y. Kim, and M.C. Keller, *Estimation of Parental Effects Using Polygenic Scores*. *Behav Genet*, 2021.
8. Robinson, M.R., et al., *Genetic evidence of assortative mating in humans*. *Nature Human Behaviour*, 2017. **1**(1): p. 0016.

Reviewers' Comments:

Reviewer #1:

Remarks to the Author:

I thank the authors for convincingly addressing all my concerns.

I have no further comments.

Reviewer #2:

Remarks to the Author:

I thank the authors for their careful and thorough work in the revision. They have sufficiently covered all the issues I raised in my last review. Congratulations on a very nice paper.

REVIEWERS' COMMENTS

Reviewer #1 (Remarks to the Author):

I thank the authors for convincingly addressing all my concerns.

I have no further comments.

Reviewer #2 (Remarks to the Author):

I thank the authors for their careful and thorough work in the revision. They have sufficiently covered all the issues I raised in my last review. Congratulations on a very nice paper.

RESPONSE

We thank the reviewers.